# QuasAr Odyssey: the origin of fluorescence and its voltage sensitivity in microbial rhodopsins

**Arita Silapetere** [1,9] ✉, **Songhwan Hwang** [1,2,9], **Yusaku Hontani** [3,8], **Rodrigo G. Fernandez Lahore** [1], **Jens Balke** [4], **Francisco Velazquez Escobar** [5], **Martijn Tros** [3], **Patrick E. Konold** [3], **Rainer Matis** [6], **Roberta Croce** [3], **Peter J. Walla** [6], **Peter Hildebrandt** [5], **Ulrike Alexiev** [4], **John T. M. Kennis** [3], **Han Sun** [2,7], **Tillmann Utesch** [2] ✉ & **Peter Hegemann** [1]

Rhodopsins had long been considered non-fluorescent until a peculiar voltage-sensitive fluorescence was reported for archaerhodopsin-3 (Arch3) derivatives. These proteins named QuasArs have been used for imaging membrane voltage changes in cell cultures and small animals. However due to the low fluorescence intensity, these constructs require use of much higher light intensity than other optogenetic tools. To develop the next generation of sensors, it is indispensable to first understand the molecular basis of the fluorescence and its modulation by the membrane voltage. Based on spectroscopic studies of fluorescent Arch3 derivatives, we propose a unique photo-reaction scheme with extended excited-state lifetimes and inefficient photoisomerization. Molecular dynamics simulations of Arch3, of the Arch3 fluorescent derivative Archon1, and of several its mutants have revealed different voltage-dependent changes of the hydrogen-bonding networks including the protonated retinal Schiff-base and adjacent residues. Experimental observations suggest that under negative voltage, these changes modulate retinal Schiff base deprotonation and promote a decrease in the populations of fluorescent species. Finally, we identified molecular constraints that further improve fluorescence quantum yield and voltage sensitivity.

Microbial rhodopsin-based voltage indicators are predominantly based on archaerhodopsin-3 (Arch3), which is a light-driven proton pump found in *Halorubrum sodomense*[1]. As a characteristic of microbial rhodopsins, Arch3 is comprised of seven transmembrane domains and a retinal chromophore that is covalently bound via retinal Schiff base (RSB) linkage. Arch3-based fluorescent voltage sensors combined with rhodopsin-based optogenetic actuators for neuron depolarization and/or hyperpolarization would allow non-invasive monitoring of membrane voltage dynamics[2], providing cell-specific all-optical electrophysiology.

The main objective of neuroscience is to understand how endogenous neuronal electrical circuits modulate brain function. It has only

[1]Institute of Biology, Experimental Biophysics, Humboldt-Universität zu Berlin, Berlin, Germany. [2]Leibniz-Institut für Molekulare Pharmakologie, Berlin, Germany. [3]Department of Physics and Astronomy, Biophysics of Photosynthesis, Vrije Universiteit Amsterdam, Amsterdam, Netherlands. [4]Department of Physics, Molecular Biophysics and Nanomedicine, Freie Universität Berlin, Berlin, Germany. [5]Department of Chemistry, Physical chemistry/Biophysical Chemistry, Technische Universität Berlin, Berlin, Germany. [6]Institute of Physical and Theoretical Chemistry, Technische Universität Braunschweig, Braunschweig, Germany. [7]Department of Chemistry, Technische Universität Berlin, Berlin, Germany. [8]Present address: Faculties of Medicine and Science, Brain Research Institute, University of Zurich, Zurich, Switzerland. [9]These authors contributed equally: Arita Silapetere, Songhwan Hwang. ✉e-mail: arita.silapetere@hu-berlin.de; utesch@fmp-berlin.de

recently become possible to use genetically encoded voltage indicators (GEVI) to track these neuronal electrical signals[3]. This is possible due to advancements in the design of GEVIs combined with modern imaging techniques. GEVIs report alterations of the membrane voltage as a change of fluorescence intensity and their expression can be targeted to selected neuronal cell types. Imaging of the fluctuations in their fluorescence signal allows simultaneous monitoring of neuron electrical activity ranging from large neuronal networks down to single cells and even to the faint signals of the dendritic spines[4]. GEVIs can be classified into three main groups: (1) GEVIs based on the fusion of voltage-sensing domains (VSD) and fluorescent proteins, (2) opsin-based GEVIs, or (3) dye-based hybrid GEVIs[1,5–16].

In this study, we focus on the opsin-based fluorescent voltage indicators, more particularly, constructs that are simultaneously voltage sensors and fluorescent indicators in a single protein. Multiple Arch3-based constructs have been reported in pursuit of improved voltage indicators (Supplementary Table 1). Although combinations of many mutations have been explored, the highest reported fluorescence quantum yield (QY) of the rhodopsin-based voltage sensor is still only ~1.2%[17]. However, the highest reported rhodopsin fluorescence QY reaches 20%[18]. Therefore, it is highly desirable to further improve the QY of rhodopsin-based voltage sensors for applications in living rodents. Difficulties can arise in the rational design of fluorescent proteins since the existing constructs were selected by screening libraries with randomly mutated Arch3. To further improve the current versions of opsin-based GEVIs, it is crucial to understand why the existing variants are fluorescent and how the voltage is controlling the fluorescence intensity.

A combination of spectroscopy studies, imaging techniques, and atomistic molecular dynamics (MD) simulations were applied here to study QuasAr1[2], QuasAr2[2], Archon1[12], and its mutants. Based on the spectroscopic studies, we proposed a photo-reaction scheme for the fluorescent proteins. Following the MD simulations, fluorescence imaging, and spectroscopy studies, we obtained a better understanding of the impact of the transmembrane voltage on protein dynamics and fluorescence. Moreover, we were able to identify the essential residues responsible for the fluorescence QY increase and voltage sensing.

## Results

### Photocycle

To study the photocycle of the microbial rhodopsin-based fluorescent voltage indicators, we chose the first generation of the highly promising constructs QuasAr1 and QuasAr2[2]. As aforementioned, QuasArs are based on the proton pump Arch3, and they carry five amino acid substitutions with respect to the wild-type (wt) (Supplementary Fig. 1a). The two constructs differ from each other with one mutation at the RSBH+ (protonated retinal Schiff base) counterion position D95H/Q, which results in different properties. The D95H in QuasAr1 replaces the negatively charged residue of the wt with histidine, granting a higher fluorescence QY and a fast, but only moderate, response to voltage changes. The D95Q in QuasAr2 replaces the negatively charged residue with neutral glutamine, causing stronger voltage sensitivity. The fluorescence QY of the parental Arch3 is estimated to be ~0.01%[17]. However, in analogy to bacteriorhodopsin (BR)[19,20], it was proposed that a higher fluorescence in wt Arch3 is reached through a three-photon process[21]. The first photon initiates a photocycle typical for microbial rhodopsins. The second photon absorbed by the N-state initiates photo-branching and brings the molecule to the fluorescent Q-state (Supplementary Fig. 2). Ultimately, the third photon absorbed by the Q-state is re-emitted[21]. The QY of the Q state in BR is estimated to be ~0.7%[19,20].

For QuasArs, it has been shown that the fluorescence is initiated through a one-photon process[2], and it remained unclear whether the bona fide dark state is fluorescent or whether equilibrium between the dark state and a fluorescent Q-like state is modulated upon voltage

application. To understand the unconventional photoresponse of the QuasArs, we studied the photoreactions using a variety of methods, such as steady-state ultraviolet–visible (UV–Vis) spectroscopy, time-resolved pump–probe spectroscopy, and retinal isomer composition analysis.

The absorption maxima of the initial dark state (IDS) of QuasAr1 and QuasAr2 are 580 nm (D580) and 590 nm (D590) (Supplementary Fig. 3a). However, both fluorescence excitation spectra match each other closely with maxima at 585 and 587 nm (Supplementary Fig. 3b, c). Additionally, we compared the one and two-photon excitation spectra for QuasAr2, and they appear to be highly similar (Fig. 1a). Upon illumination of QuasAr2 D590 with a 625 nm LED for 60 s, we first observed a 10 nm hypsochromic shift of the absorption maxima to P580, which was then followed by deprotonation and accumulation of the UV absorbing photoproduct P400 (Fig. 1b, c). Both the P580 and P400 species are thermally stable, and the D590 does not recover during 24 h (Supplementary Fig. 3d)[22,23]. The P400 photoproduct is also photoactive, and it was found that upon 60 s blue illumination (400 nm), the dark state-like D590' is recovered (Fig. 1b, c). The steady-state UV–Vis spectra indicate that the QuasArs populate three subsequent photoproducts that are thermally stable. Furthermore, it is possible to switch between the protonated and deprotonated species upon illumination. Similar behavior has been observed for QuasAr1[23]. However, the transient absorption spectra (TAS), recorded in the time range between 10 ns and 10 s after a 5 ns laser flash, did not show any evidence of photointermediate formation (Supplementary Fig. 3e). This implies that the transitions between the photoproducts are too inefficient to record spectral changes in single flash experiments.

To study the excited-state dynamics of QuasArs, we employed femtosecond pump-probe spectroscopy. This method allows tracking of absorption changes after photoexcitation in a time range from femtoseconds to microseconds. TAS of QuasAr2 was recorded in the spectral region of 360–720 nm with excitation at 620 nm and is shown in Fig. 1d–f. The fs–μs TAS of QuasAr1 is shown in Supplementary Fig. 4a–c. The positive peak around 510 nm is assigned to excited-state absorption (ESA), and the negative bands at ~590 and >700 nm are attributed to ground state bleaching (GSB) and stimulated emission (SE), respectively[24,25].

The TAS of QuasAr2 was globally fitted with five exponential constants, i.e., 183 fs ($\tau_0$), 4 ps ($\tau_1$), 40 ps ($\tau_2$), 288 ps ($\tau_3$), and an infinite time component ($\tau_4$) (Fig. 1g). Upon photoexcitation, D590 of QuarAr2 is excited to the $S_1$ state. Frank–Condon relaxation takes place within 183 fs and a negative band of SE at >700 nm emerges. The excited-state relaxation with the SE signals decays within 4 and 40 ps. The excited-state lifetime of QuasAr2 is significantly longer than that of the Arch3 (~0.3 ps) (Supplementary Fig. 4d–i) immediately explaining the increased fluorescence.

Along with the excited-state decay, another positive band gradually appears at around 640 nm in the picosecond time scale. This small positive band in Fig. 1e, f is assigned to the isomerized photo-intermediate I640, which evolves to I620 within 288 ns. The decay time of I620 is beyond the time range of this measurement (i.e., 1 μs). Although a large fraction of molecules is excited (designated by a strong signal in TAS), only a minor fraction of the excited chromophore isomerizes. QuasAr2 and QuasAr1 exhibit small isomerization quantum yields that are estimated to be ~4% and ~1%, respectively.

In QuasAr2, the shorter-lived ($\tau_1$) and the longer-lived species ($\tau_2$) nearly equally contribute to the excited-state decay; but in QuasAr1, the contribution of the longer-lived component ($\tau_2$) is ~90% (Fig. 1e, h, Supplementary Fig. 4). Moreover, the time constant of the longer-lived decay component of QuarAr1 is 1.5-fold larger than that of QuasAr2 (60 vs. 40 ps). These factors contribute to the 1.6-fold higher fluorescence QY of QuasAr1 (Fig. 1h, Supplementary Fig. 3f).

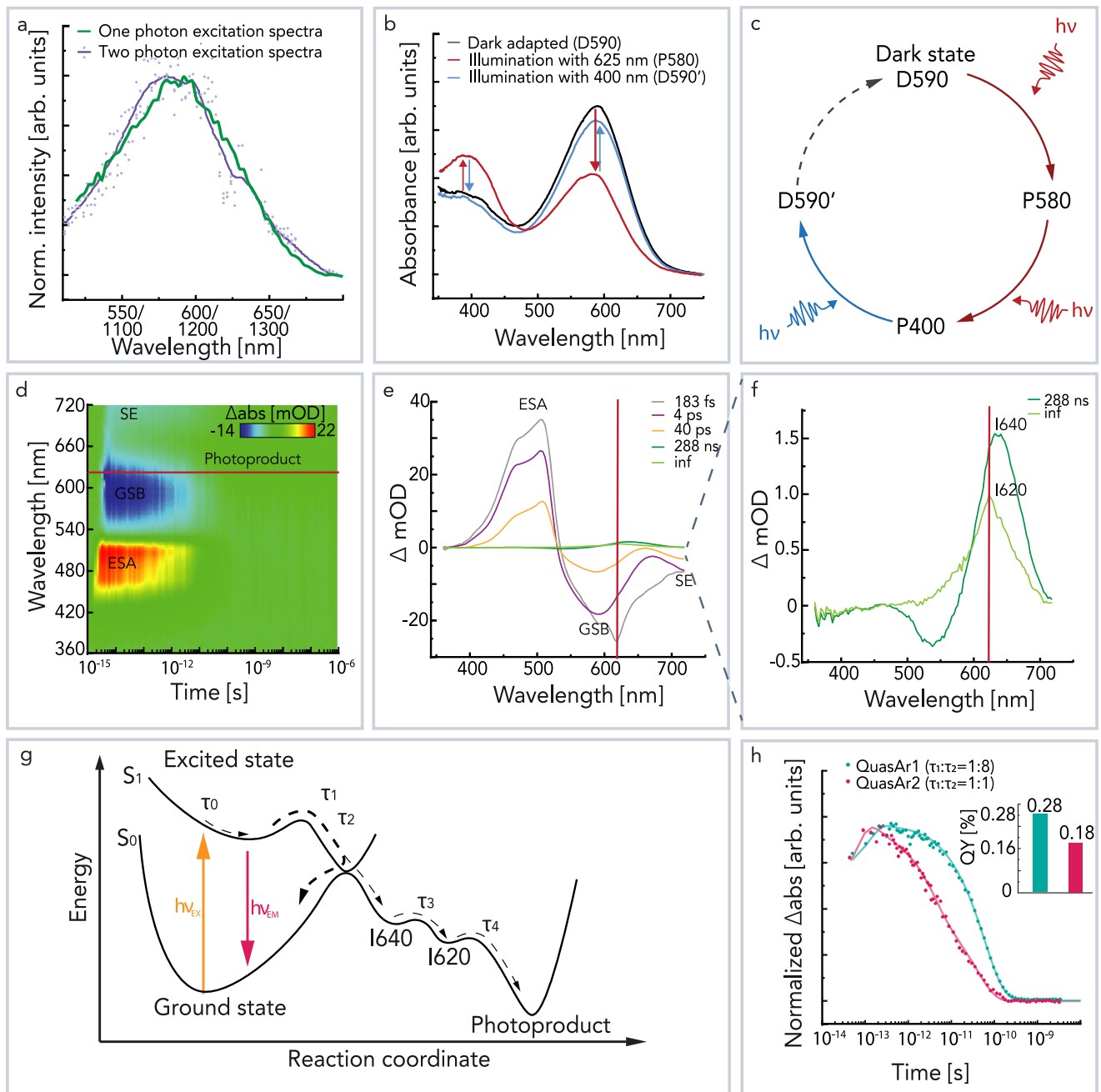

**Fig. 1 | Study of the photocycle of QuasArs. a** Excitation spectra of QuasAr2 compared between one-photon (green) and two-photon (purple) excitation. The two-photon excitation spectra data points are depicted with purple dots, and the resulting average is indicated with a solid line for visual guidance. The corresponding excitation wavelengths for two-photon spectra are noted below. **b** The QuasAr2 absorption spectra change after illumination with LED. The black curve shows IDS D590 absorption. The red curve shows a mixture of P580 and P400 photoproduct absorption spectra after sample illumination for 60 s with 625 nm. The blue curve shows a dark state-like recovered D590′ photoproduct absorption after subsequent 60 s 400 nm illumination. **c** Schematic representation of photoproduct formation after illumination. **d**–**f** The fs-to-μs transient absorption spectra of QuasAr2 obtained with fs pump-probe spectroscopy are shown in the middle row, where **d** reconstructed 3D plot of TAS, **e** evolution-associated difference spectra (EADS), and **f** scaled-up EADS. **g** The reaction coordinate of the excited-state dynamics is shown. Upon light excitation the QuasAr2 is brought to the excited-state $S_1$, followed by Franck–Condon decay in $\tau_0$. The excited state decays with two-time constants—$\tau_1$ and $\tau_2$. Minor (~4%) fraction of the excited molecules undergo isomerization and contribute to the formation of a photointermediate I640, which transitions into photointermediate I620 with $\tau_3$ and then decays with $\tau_4$. After excitation, most of them relax to the initial ground state. The excited state is long-lived and results in slow decaying SE. **h** Comparison of normalized excited-state decay between QuasAr1 and QuasAr2. The subpanel indicates the fluorescence QY determined in this study of the QuasArs, using the same excitation wavelengths as for pump-probe experiments (620 nm). The spectra of the purified sample were recorded in room temperature (RT) and buffer solution at pH 8. Source data are provided in the Source Data file (**a**–**h**).

To further investigate QuasAr1 and QuasAr2, we studied the retinal isomer composition by hexane extraction and high-performance liquid chromatography (HPLC). The IDS of QuasAr1 is composed of 21% 13-*cis* and 79% all-*trans* (Fig. 2a), whereas the IDS of QuasAr2 is composed of 47% 13-*cis* and 53% all-*trans* (Fig. 2b).

These ratios relate to the heterogenous $S_1$ decay and the relative contribution of the $\tau_1$ and $\tau_2$ components. The dark state retinal isomer composition was confirmed by pre-resonance Raman spectroscopy (Supplementary Fig. 5). The photoproduct retinal isomer compositions of QuasAr2 are shown in panels Fig. 2c, d. The mixture

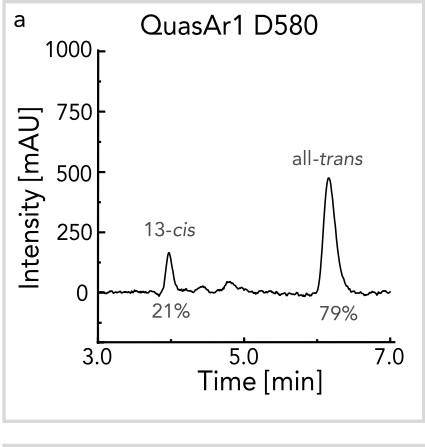

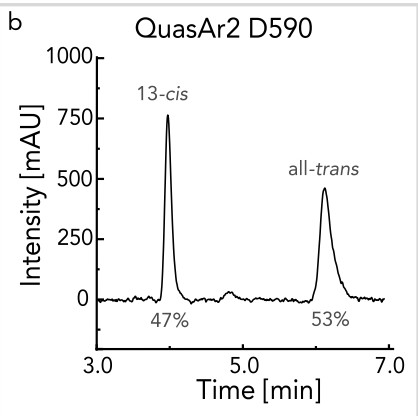

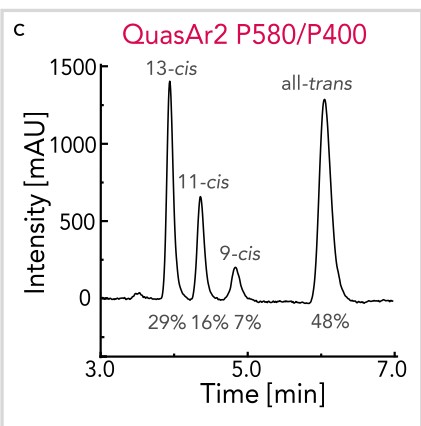

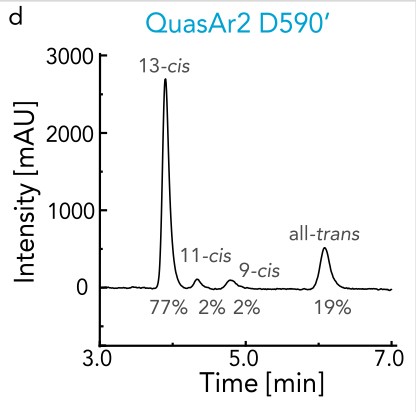

**Fig. 2 | Retinal extract HPLC chromatogram.** HPLC of extracted retinal isomers isolated from IDS of QuasAr1 (**a**) and QuasAr2 (**b**). For QuasAr2, the photoproduct chromatograms are shown for P580/P400 (**c**) and D590′ (**d**). The sample was illuminated for 30 s with 625 nm LED (**c**) and additional 30 s illumination with 400 nm LED (**d**) before the retinal extraction. The ratio of eluted retinal isomers was determined via peak integration and is shown as a percentage in the bottom part of the image. Source data are provided in the Source Data file.

of P580 and P400 states showed nearly no change in all-*trans* (48%) isomer contribution but exhibited an increase in 11-*cis* (16%) and 9-*cis* (7%) retinal isomers at the expense of the 13-*cis* (29%) retinal isomer. The contribution of 11-*cis* and 9-*cis* retinal isomers was unexpected, as these isomers are rarely found in microbial rhodopsins. The dark state-like QuasAr2 D590′ photoproduct mainly showed contributions from 13-*cis* (77%), decreased all-*trans* (19%) isomer, and only traces of 11-*cis* (2%) and 9-*cis* (2%) despite the absorption being almost identical to the IDS (Fig. 1b). The variety of the retinal isomers detected in the photoproducts reveal the heterogeneous nature of the QuasArs. The fluorescence QY slightly differs between the IDS and photoproducts (Supplementary Fig. 6, Supplementary Table 2), where the sub-states with all-*trans* chromophores are more fluorescent than the 13-*cis*.

**Voltage-dependent fluorescence**

To understand the functionality of fluorescent microbial rhodopsins, it is crucial to analyze how the membrane voltage is controlling the fluorescence intensity. Here, we discuss multiple possible mechanisms. Membrane voltage changes may affect (1) the hydrogen bonding network and the rigidity of the retinal binding pocket, (2) the equilibrium of retinal isomers 13-*cis* and all-*trans*, (3) the protonation of the RSB or adjacent charged residues, and (4) the fluorescence lifetime and QY.

We selected Archon1 as the model construct owing to its further improved fluorescence intensity, voltage sensitivity, and membrane targeting[12]. Archon1 is based on QuasAr2, and conserves the key mutations at the retinal binding pocket, in particular the counterion mutation D95Q (Supplementary Fig. 1b). The excited-state dynamics of Archon1 (Supplementary Fig. 7) are similar to the dynamics of QuasAr2 (Fig. 1e), assuring the compatibility of the developed models. In the case of Archon1, the excited-state lifetime is even longer (Supplementary Fig. 7), and it decays with two-time components $\tau_1 = 14$ ps and $\tau_2 = 75$ ps. Furthermore, the isomerization efficiency decreased from ~4% to ~0.1%.

Atomistic MD simulations were carried out to gain insights into the voltage-induced dynamics of the wt Arch3, Archon1, and its variants on the atomic level. The MD simulations were performed on corresponding homology models, based on the high-resolution X-ray structure of Arch2 (sequence identity of 80%) harboring a protonated all-*trans* retinal. The homology models were in close agreement with predictions from Colab AlphaFold2[26,27] and RoseTTAFold[28] (Supplementary Fig. 8). Already previously, classical MD simulations and quantum mechanics/molecular mechanics (QM/MM) simulations have been used to study the role of the hydrogen bonding network in microbial rhodopsins for ion transport[29-31], but their involvement in voltage-sensitive fluorescence remained elusive. In this study, we investigated the voltage sensitivity of Archon1 and its mutants at a wide range of positive and negative transmembrane voltages (up to ca. ±900 mV) (Supplementary Fig. 9) using the computational electrophysiology approach[32]. Higher voltages compared to physiology-relevant conditions were applied to accelerate voltage-induced conformational changes in the ns–μs time scale, which may occur at physiological conditions only on slower time scales. However, it should be noted that these high transmembrane voltages did not significantly affect the overall structural integrity of the proteins during the current simulation time. Each MD trajectory undergoes only minor structural changes to reach its own equilibrium state of the protein regardless of the applied transmembrane voltage. This is

demonstrated by the root mean square deviations (RMSDs) analysis of the protein backbone (Supplementary Fig. 10) and the comparison of the final snapshots obtained from the simulations performed at different transmembrane voltages (Supplementary Fig. 11). Here, the proteins behave similarly at high voltages and 0 mV. Differences in the protein are, however, visible in the orientation of the side chains and consequently in the hydrogen bonding network as discussed in the following. Additionally, no electroporation was observed in our simulations.

The wt Arch3 model is also based on the high-resolution structure of Arch2[33,34], and the integrity of the model was validated with the recently published X-ray structure of Arch3[29] (Supplementary Fig. 12). The MD simulations of Arch3 identified the highly conserved R92 as the key residue for voltage sensing. The homolog R82 in BR has been reported to switch between inward and outward orientations during the photocycle[35]. Our MD simulations predicted that at negative voltage R92 is pointing towards the RSBH$^+$ and strongly interacted with the adjacent negatively charged counterions D95 and D222 (Supplementary Fig. 13a, d). However, at positive voltages, the side-chain of R92 reoriented towards the extracellular space and formed new stable salt bridges with the proton release pair E204 and E214 (Supplementary Fig. 13b, d). The conformational changes of R92 prohibited water molecules from entering the vicinity of the RSBH$^+$ (Supplementary Fig. 14). Furthermore, under positive voltage a hydrogen bond between D95 and T99 (HB1) was established (Supplementary Fig. 13b, e), leading to more rigid D95 and decreased the interaction between D95 and the RSBH$^+$.

In contrast to the homology model of Arch3, the R92 flipping was not observed in the simulations of Archon1. The removed negative charge at the counterion position in Archon1 (D95Q) resulted in the loss of the electrostatic interaction between the positively charged R92 and neutral Q95. Independent of the applied voltage, R92 faced the extracellular space and interacted with E204 and E214 in all simulations (Fig. 3a, b and Supplementary Fig. 13c, d). Instead, a voltage-dependent reorientation of intracellular D125 (helix D) was predicted (Fig. 3a, b, Supplementary Fig. 13e, Supplementary Fig. 15). The MD simulations indicate that movement of D125 originates from the overall increased flexibility of the intracellular part of helix D (Supplementary Fig. 15b–d). At negative voltage, the carboxyl side-chain of D125 pointed towards the retinal, forming a hydrogen bond with the hydroxyl side-chain of T100 (helix C) (Fig. 3a, e, HB3). At positive voltage, the simulations displayed flipping of D125 towards the intracellular space, disrupting HB3. D125 establishes a new hydrogen bond with the backbone NH of adjacent G123 located in helix D (Fig. 3b, e, HB4). This, in turn, led to a significant rearrangement of the hydrogen bonding network around the chromophore, in particular the TT motif (T99 and T100). The side-chain of T100 interacted with the backbone of W96 (HB2) and the side-chain of T99 with the side-chain of Q95 (HB1). The formation of HB1 pulls Q95 away from the RSBH$^+$ (Fig. 3d, e). Consequently, at positive voltages, hydrogen bonding between Q95 and the RSBH$^+$ became less probable, but the interaction between D222 and the RSHB$^+$ strengthened (Fig. 3d). Additionally, the observed narrower distance and angle distributions between D222–RSBH$^+$ and Q95–RSBH$^+$ suggest a higher rigidity of the chromophore environment (Supplementary Fig. 16).

The counterion mutation D95Q is associated with another significant observation in MD simulations. In Archon1 the Q95 side-chain formed a stable contact with D222 (Fig. 3d), which further increased the rigidity of the region around the RSBH$^+$ and eliminated the water from this region. In contrast, the parental Arch3 showed voltage-dependent water influx (Supplementary Fig. 14). Presumably the predominant hydrogen bond between D222 and RSBH$^+$ also yields the calculated low and voltage-independent p$K_a$ of ~3 of D222 (Supplementary Fig. 17).

## Archon1 variants

To confirm the role of essential residues predicted by MD simulations to be involved in voltage sensing, we carried out further experimental studies and MD simulations of selected Archon1 variants. First, we studied the significance of R92 and substituted this residue with equally charged and less bulky Lys (Fig. 3c).

Both MD simulations and experimental results suggested Archon1-R92K as less voltage-sensitive compared to Archon1. The MD simulations predicted that also K92, like R92, is oriented towards the extracellular side and interacts with E204 and E214 independent of membrane voltage (Fig. 3c, Supplementary Fig. 18a). Additionally, the simulations revealed almost no dependence on the hydrogen bonding network on the transmembrane voltage (Fig. 3e, Supplementary Fig. 19a). This includes the stable hydrogen bond between the RSBH$^+$ and D222 (Fig. 3d) and the interactions of Q95, in particular with D222 (Supplementary Figs. 16 and 21) and to a much weaker content to T99 (HB1). Only at a high negative voltage, the MD simulations display disruption of the hydrogen bond between Q95 and D222 (Supplementary Fig. 21) and consequent water influx in the retinal binding pocket, near the RSBH$^+$ (Supplementary Fig. 23a). This suggests that R92 and the Q95–D222 contact in Archon1 play a role of 'gatekeepers', prohibiting water protrusion in the chromophore cavity (Supplementary Fig. 14e–h).

To test the voltage-sensitive fluorescence, the variants were expressed in ND7/23 cells. The fluorescence intensity change ($\Delta F/F$, %) was compared in whole cells voltage clamp experiments (20 mV steps from −80 to 40 mV) (Fig. 3f). The voltage sensitivity decreased upon R92K replacement from $\Delta F/F_{max} \approx 20\%$ to $\Delta F/F_{max} \approx 7\%$.

The purified Archon1-R92K displayed similar absorption maxima as Archon1 (585 nm), but it had a decreased retinal stability, as indicated by an increased $A_{280}/A_{580}$ ratio (Fig. 3g, h). Interestingly, the Archon1-R92K variant showed higher (0.74%) fluorescence QY than Archon1 (0.49%), which could indicate higher retinal rigidity (Fig. 3i). The observations in MD simulations support that the rigidity of chromophore at 0 mV in Archon1-R92K compared to Archon1 is indeed increased (Fig. 3d). To explore the correlation of the voltage sensitivity and the p$K_a$ of the RSB, pH titration of the RSB was carried out with purified variants (Fig. 3g, h). Despite the different fluorescence voltage responses, the p$K_a$ ~ 9 of Archon1 is conserved in Archon1-R92K.

Based on the results from the MD simulations, we studied the role of T100 and D125 in the voltage sensing mechanism in the next step (Fig. 4a). The fluorescence intensity change ($\Delta F/F$, %) of these mutants upon membrane voltage variation is shown in Fig. 4b. The constructs Archon1-T100A and Archon1-T100S show different impacts on the voltage-sensitivity. The reduction of the side chain to a methyl group in Archon1-T100A makes hydrogen bonding impossible and results in a decreased voltage sensitivity ($\Delta F/F_{max} \approx 12\%$). In contrast, the Archon1-T100S variant, with a hydroxymethyl side-chain conserving the chance to form a hydrogen bond with reduced steric hindrance, showed similar or even slightly increased voltage sensitivity ($\Delta F/F_{max} \approx 24\%$) compared to Archon1. Archon1-T100S, just as parental Archon1, showed excitation wavelength-dependent voltage sensitivity with higher $\Delta F/F_{max}$ upon increasing excitation wavelength (620 nm) (Fig. 4c). Furthermore, the $\Delta F/F_{max}$ slope is not linear and depends on the selected membrane voltage range. The variant Archon1-T100S showed a faster increase in fluorescence intensity in the application relevant membrane voltage range −60 to 0 mV with 560 nm excitation, but 620 nm excitation showed the highest fluorescence increase in the experimental range from −80 to +40 mV. These results were supported by MD simulations, which showed a strong effect of these mutations on the voltage-dependent hydrogen bonding network (Fig. 4d, e and Supplementary Fig. 19b, c) ranging from HB1 (Q95-T99) to HB5 (D125-

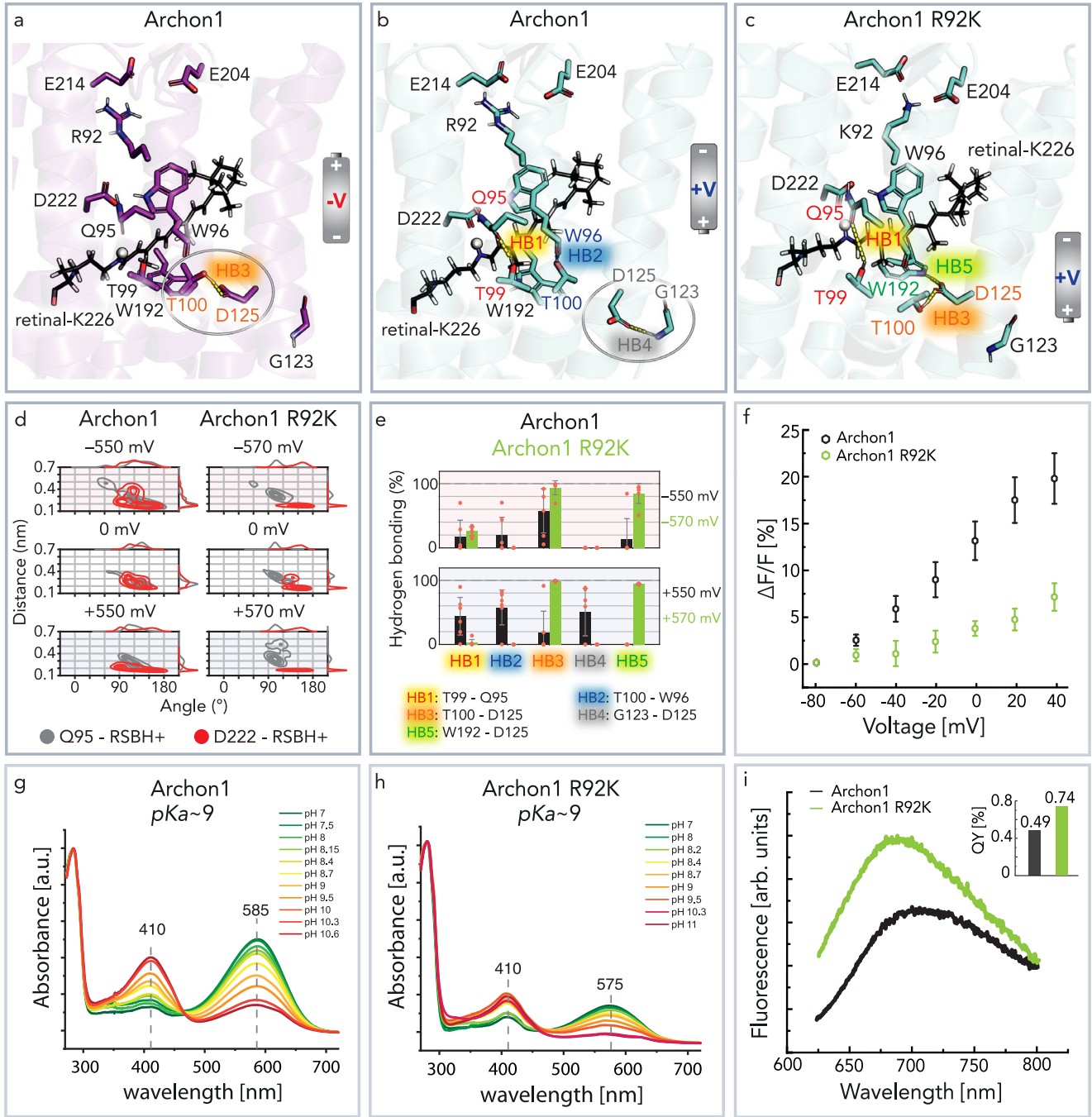

**Fig. 3 | Voltage sensing in Archon1. a**, **b** Voltage-induced conformations of Archon1 as predicted by MD simulations. **a** At negative voltage, the key residue D125 interacts with T100 by hydrogen bonding (HB3). **b** This contact breaks at the positive voltage and D125 flips towards intracellular G123 (HB4). The voltage-induced side-chain reorientation strongly affects the complete hydrogen bonding network (from HB1 to HB4) around the retinal (retinal: black stick representation; Schiff base proton: white sphere). **c** Hydrogen bonding network as observed by MD simulations at a positive transmembrane voltage in Archon1-R92K. **d** Distance and angle distributions between the RSBH⁺ and the two counterions Q95 (gray) and D222 (red) as observed by MD simulations at positive, zero, or negative trans-membrane voltage in Archon1 and Archon1-R92K. **e** Probability of key hydrogen bonds in Archon1-R92K (green) compared to Archon1 (black) at positive, zero, or negative transmembrane voltage (the number of simulations for each voltage = 6). All listed hydrogen bonds are established between the side chains. Only W96-T100

and G123-D125 interactions are established via the backbone. HB5 is depicted in Supplementary Fig. 13c. Data are presented as mean values ± SD. Each orange circle represents the data derived from the individual MD trajectories. **a**–**e** The MD simulations were conducted at 303 K by fixing the protonation states of titratable sites at neutral pH. **f** Voltage-dependent increase in fluorescence intensity ($\Delta F/F$) under continuous 620 nm excitation (Archon1 $n = 14$; Archon1-R92K $n = 4$, $p$-value 0.017, the $p$-value is determined according to Wilcoxon–Mann–Whitney test (two-sided), Supplementary Fig. 22, Supplementary Table 3). Recordings were carried out in ND7/23 cells under patch-clamp voltage control at RT and pH 7.2. **g** pH titration of RSB of Archon1. **h** pH titration of RSB of Archon1-R92K. The absorption spectra maximum of Archon1 (585 nm) and Archon1-R92K (575 nm) are compared at neutral pH values. **i** Fluorescence emission and QY comparison between Archon1 and Archon1-R92K. The spectra were recorded at RT and pH 8.0. Source data are provided in the Source Data file (**d**–**i**).

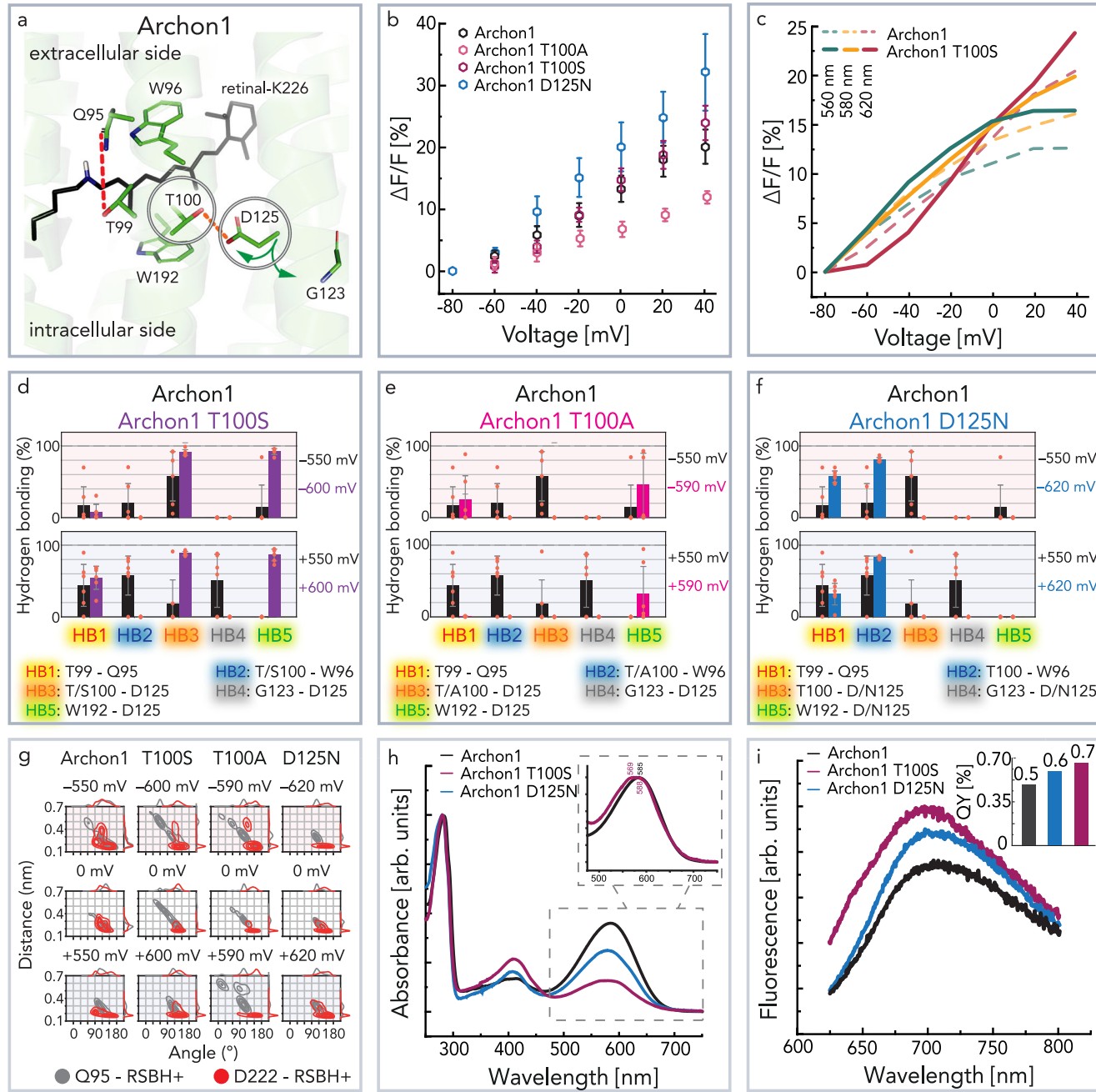

**Fig. 4 | Archon1-T100S/A and Archon1-D125N. a** Retinal binding pocket high-lighting mutated residues−T100 and D125. **b** Voltage-dependent increase in fluorescence intensity of Archon1, Archon1-T100A, Archon1-T100S, Archon1-D125N at continuous 620 nm excitation RT, pH 7.2 (Archon $n = 14$; Archon1-T100A $n = 4$, $p$-value 0.184; Archon1-T100S $n = 4$, $p$-value 0.367; Archon1-D125N $n = 4$, $p$-value 0.080, the $p$-values are determined according to Wilcoxon−Mann−Whitney test (two-sided), Supplementary Fig. 22, Supplementary Table 3). **c** Comparison between Archon1 (dashed lines) and Archon1-T100S (solid lines) voltage regulated fluorescence intensity (Δ$F/F$) dependent on excitation wavelength −560 nm (green, $p$-value 0.327), 580 nm (orange, $p$-value 0.313), 620 nm (red) ($n = 4$, $p$-value 0.367, the $p$-values are determined according to Wilcoxon−Mann−Whitney test (two-sided), Supplementary Fig. 22, Supplementary Table 3), measured at pH 7.2 and RT.

**d**–**f** Voltage-dependent probability of important hydrogen bonds in T100S (purple), T100A (pink), and D125N (blue) compared to Archon1 (black) using the same nomenclature as in Fig. 3 (the number of simulations for each voltage = 6). Data are presented as mean values ± SD. Each orange circle represents the data derived from the individual MD trajectories. **g** Distance and angle distributions between the RSBH⁺ and the two counterions Q95 (gray) and D222 (red) as observed by MD simulations at negative, zero, and positive transmembrane voltages in Archon1, Archon1-T100S, Archon1-T100A, and Archon1-D125N. **d**–**g** The MD simulations were conducted at 303 K by fixing the protonation states of titratable sites at neutral pH. **h** Absorption spectra comparing Archon1, Archon1-T100S, and Archon1-D125N. **i** Fluorescence emission and QY. Spectra of panels **h** and **i** have been recorded at RT, pH 8.0. Source data are provided in the Source Data file (**b**–**i**).

W192). In contrast to Archon1-T100A, we predicted for Archon1-T100S a voltage-insensitive formation of HB3 between the side chains of S100 and D125. More importantly, Archon1-T100S exhibited a voltage-dependent interaction between Q95 and T99 (HB1). Surprisingly, the increased probability of HB1 at positive voltage increased also the probability of hydrogen bonding between Q95 and the RSBH⁺(Fig. 4g). As observed for Archon1, the retinal binding pocket including the RSBH⁺ became more structurally rigid with increasing voltages (Fig. 4g). In particular, Q95 was affected. In agreement with all D95Q mutants, very seldomly water molecules entered between the two counterions (Supplementary Figs. 21 and 23b, c).

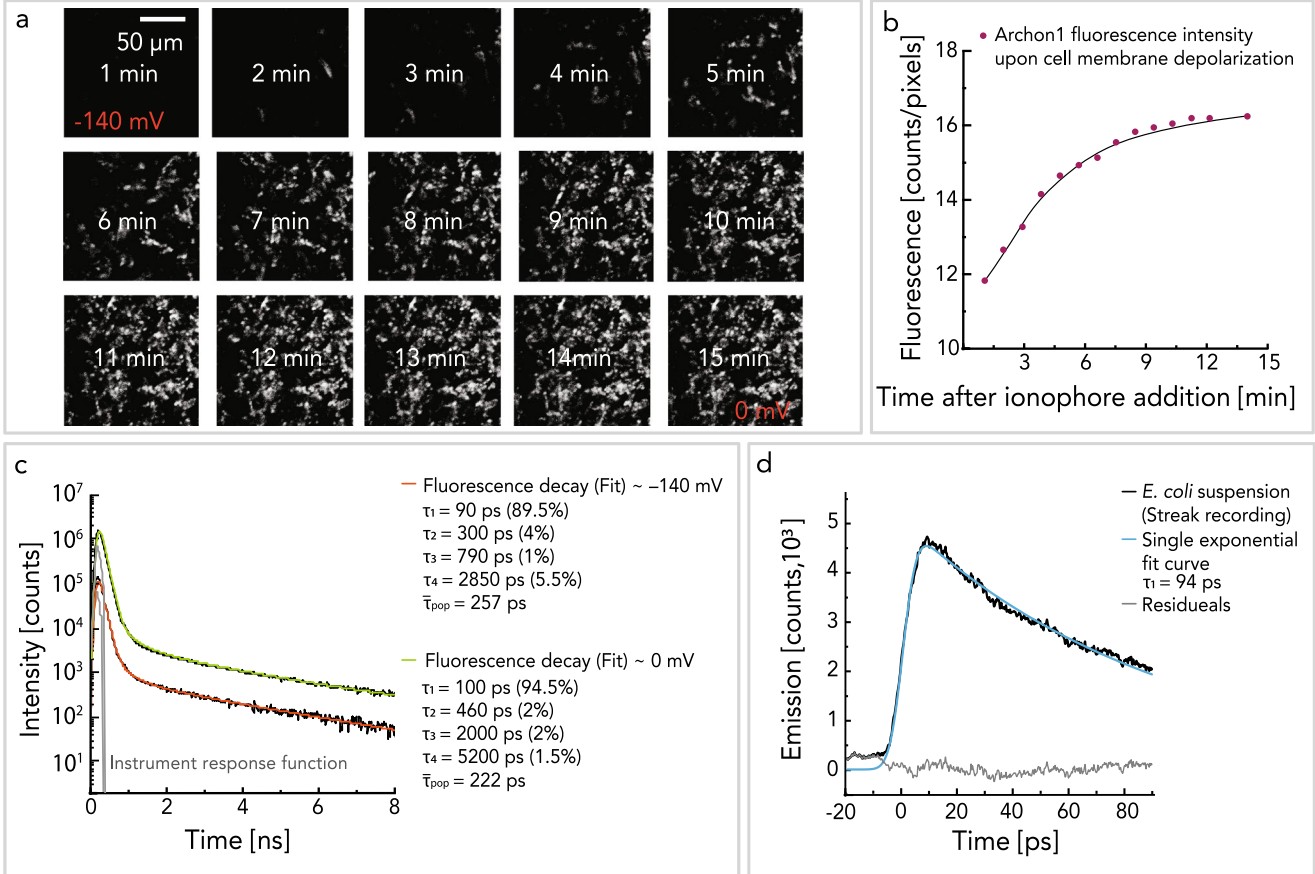

**Fig. 5 | Fluorescent lifetime measured in cells by FLIM. a** 15 subsequent images of *E. coli* cell suspension with overexpressed Archon1 after the addition of gramicidin ionophore. Each image was recorded for 1 min. The resting voltage of *E. coli* cells is reported to be ~−140 mV[36], which after the addition of gramicidin collapses to 0 mV. The fluorescence decay of single *E. coli* cells was measured at RT, pH 8.0. The excitation pulse (~60 ps) was tuned to 640 nm and the time resolution was 19.8 ps. **b** The Archon1 fluorescence intensity upon membrane depolarization was extracted from the images in panel (**a**). The fluorescence intensity exhibited by the *E. coli* cells was normalized to the number of fluorescent pixels in the image to detect the fluorescence increase of the *E. coli* cells. The fit with a Hill-model yield a time constant of ~4 min for gramicidin-based membrane depolarization. **c** The corresponding fluorescence lifetime fits before and 15 min after the addition of the ionophore. The fit results and the mean fluorescence lifetime $\bar{\tau}_{pop}$ are given on the right side of the panel. **d** The fluorescence (after 625 nm excitation) of Archon1 overexpressed in *E. coli* suspension was recorded with a Streak camera setup (3–4 ps time resolution). The fluorescence decay could be described by a single exponential function (blue) with $\tau_1 = 94$ ps. Source data are provided in the Source Data file (**b–d**).

Surprisingly, the mutation of the voltage-sensitive residue D125 to asparagine resulted in an even further increase in the voltage sensitivity of the fluorescence ($\Delta F/F_{max} \approx 33\%$; Fig. 4b). Also Archon1-D125N variant showed excitation wavelength-dependent on voltage sensitivity ($\Delta F/F_{max}$), and the highest fluorescence intensity increase with 620 nm excitation (Supplementary Fig. 25a). Archon1-D125N revealed increased voltage sensitivity, despite its expression in ND7/23 being decreased. However, the experimentally determined high voltage sensitivity of the Archon1-D125N mutant could neither be predicted nor explained by the MD simulations. The hydrogen bonding network of the retinal binding pocket was insensitive to different transmembrane voltages (Fig. 4f, g, Supplementary Figs. 20a, and 24a). We hypothesize that Archon1-D125N uses a different voltage-sensing mechanism compared to Archon1, where current MD simulations under the ns-μs time scales and excluding other retinal isomers fail to reproduce the experimental observations.

The absorption spectra of the improved variants Archon1-T100S (580 nm) and Archon1-D125N (580 nm) are displayed in Fig. 4h with spectral broadening for Archon1-T100S designated in the sub-panel. Both mutants revealed an increased fluorescence QY in comparison to Archon1 (Fig. 4i). The pH titration of the two variants is described in supplementary Supplementary Fig. 25b,c, with both variants resulting in p$K_a$ ~ 9.

**Spectroscopy under variation of the membrane voltage**

To experimentally study the voltage-dependent fluorescence kinetics and retinal isomer composition, we established an *E. coli* cell assay with overexpressed Archon1. The resting membrane voltage of *E.coli* cells is reported to be ~−140 mV[36], which, after the addition of the ionophore gramicidin[37], should collapse close to 0 mV. Recordings of microscopic fluorescence lifetime images (FLIM) gave insights into Archon1 fluorescence lifetime changes as they occurred upon membrane depolarization. Figure 5 displays the images recorded over a period of 15 min. Upon membrane depolarization, there is a gradual increase in fluorescence up to a stationary value that is reached when the membrane voltage approaches 0 mV (Fig. 5a, b). The Archon1 excited state fluorescence decay, as observed in the live-cell FLIM experiments, can be fitted exponentially with four decay components. The fastest stimulated emission decay time component in vitro is 14 ps and is below the time resolution of this setup (50 ps)[38], but additional slower fluorescence decay components can be resolved (Supplementary Fig. 7). Before the addition of gramicidin, the membrane voltage is at the resting value (~−140 mV), and the fluorescence decay can be fitted

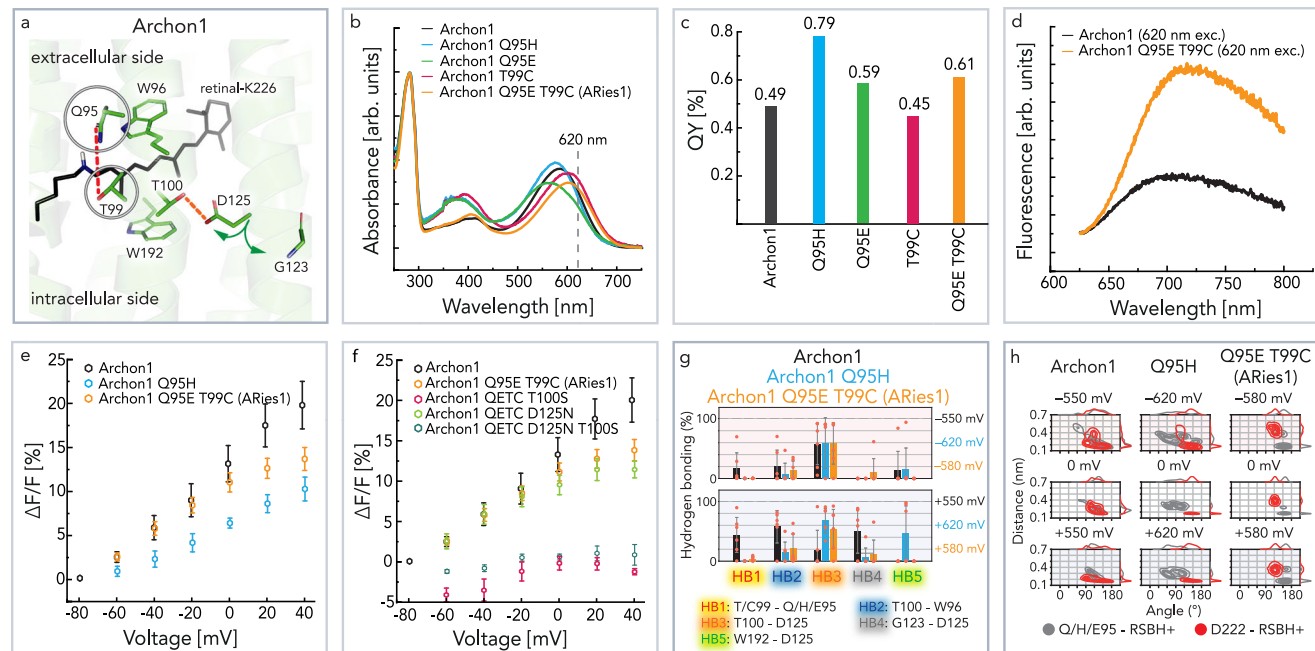

**Fig. 6 | Mutant studies of fluorescence increase. a** Retinal binding pocket highlighting mutated residues−Q95 and T99. **b** Absorption spectra of Archon1 and its D95X variants. **c** Fluorescence QY of Archon1 and its D95X variants. **d** Emission spectra of Archon1 and ARies1 with 620 nm excitation. Spectra presented in panels **b**−**d** have been recorded at RT, pH 8.0. **e** Voltage-dependent increase in fluorescence intensity of Archon1, Archon1-Q95H, and ARies1 under continuous 620 nm excitation, RT, pH 7.2 (Archon $n = 14$; Archon1-Q95H $n = 4$, $p$-value 0.049; ARies1 $n = 8$, $p$-value 0.010, the $p$-values are determined according to Wilcoxon−Mann−Whitney test (two-sided), Supplementary Fig. 22, Supplementary Table 3). **f** Voltage-dependent increase in fluorescence intensity of combinations of red-shifted and highly voltage-sensitive constructs (ARies1 $n = 8$; QETC-T100S $n = 3$, $p$-value 0.001; QETC-D125N $n = 4$, $p$-value 0.038; QETC-D125N-T100S $n = 3$, $p$-value 0.010,

the $p$-values are determined according to Wilcoxon−Mann−Whitney test (two-sided), Supplementary Fig. 22, Supplementary Table 3). **g** Voltage-dependent probability of important hydrogen bonds in Archon1-Q95H (blue) and ARies1 (orange) compared to Archon1 (black) predicted by MD simulations using the same nomenclature as in Fig. 3 (the number of simulations for each voltage = 6). Data are presented as mean values ± SD. Each orange circle represents the data derived from the individual MD trajectories. **h** Distance and angle distributions between the RSBH⁺ and the two counterions Q95 (gray) and D222 (red) as observed by MD simulations at positive, zero, and negative transmembrane voltages in Archon1, Archon1-Q95H, and ARies1. **g**, **h** The MD simulations were conducted at 303 K by fixing the protonation states of titratable sites at neutral pH. Source data are provided in the Source Data file (**b**−**h**).

with $\tau_1 = 90$ ps (89.5%), $\tau_2 = 300$ ps (4%), $\tau_3 = 790$ ps (1%), $\tau_4 = 2850$ ps (5.5%). After 15 min, the membrane voltage is expected to be close to ~0 mV, and the fluorescence decay can be fitted with $\tau_1 = 100$ ps (94.5%), $\tau_2 = 460$ ps (2%), $\tau_3 = 2000$ ps (2%), and $\tau_4 = 5200$ ps (1.5%). This leads to a similar mean fluorescence lifetime at resting membrane potential values (~260 ps) and upon depolarization (~220 ps) within the fitting error (5–10%) under live cell conditions (Fig. 5c). Additionally, in vitro recordings of Archon1, reconstituted in proteoliposomes, support our observations in the *E. coli* cell assay (Supplementary Fig. 26a–d).

To investigate fluorescence decay components shorter than ~50 ps, such as short-lived excited-state decay component of ~14 ps (Supplementary Fig. 7), we measured fluorescence decay spectra with a synchro-scan streak camera system, yielding a time resolution of 3 ps (Fig. 5d, Supplementary Fig. 27). Figure 5d shows the fluorescence decay kinetics of Archon1 overexpressed in *E. coli* cells (black line), along with an exponential fit (blue). A single fluorescence decay time component (94 ps) could be resolved in the selected time window, which corresponds to the fast time component $\tau_1$ observed with FLIM. This demonstrates that no short-lived component (i.e., 14 ps) is present in vivo Archon1.

To further analyze the voltage dependence of the constructs, we used the *E. coli* cell assay to measure changes in the chromophore structure upon membrane depolarization by pre-resonance Raman spectroscopy (Supplementary Fig. 28). The differences observed between the two spectra were minor. The C=C stretch peak frequency at 1522 cm⁻¹ remained unchanged upon membrane depolarization, indicating that a shift in absorption maxima upon membrane

depolarization is unlikely (Supplementary Fig. 28e). The fingerprint region, which is characteristic of the chromophore configuration and conformation, shows an intensity redistribution of the band pair at 1165 and 1185 cm⁻¹, with respect to the 1200 cm⁻¹ band. The 1200 cm⁻¹ band gains intensity upon membrane depolarization. This change may indicate a slight increase in all-*trans* retinal in comparison to 13-*cis* retinal upon depolarization (Supplementary Fig. 28d).

## Fluorescence modulation by protein engineering

We finally intended to study the mutations that cause the increase in fluorescent QY. Previous[1,2] studies and our studies suggest that the counterion mutation D95X (where X can be different residues) is the most fundamental one, defining the fluorescence QY, absorption maxima, and RSB p$K_a$ values (Fig. 6, Supplementary Fig. 29, Supplementary Table 4).

As shown before, the counterion mutation that distinguishes QuasAr1 (D95H) from QuasAr2 (D95Q) results in different fluorescence QY of the constructs[2]. Accordingly, also in Archon1, the substitution Q95H gives a blue shift of absorption maximum by 10 nm (Fig. 6b) and an increase in fluorescence QY to 0.79% (Fig. 6c), whereas the voltage sensitivity has decreased two-fold to $\Delta F/F_{max} \approx 10.8\%$ (Fig. 6e). The change of residue at the counterion position did not reintroduce photocurrents (Supplementary Fig. 30). Additionally, the MD simulations revealed that the introduction of a histidine diminished the voltage-dependent hydrogen bond rearrangement. In particular, the hydrogen bonding between D222 and H95 was decreased compared to the Q95 variants (Supplementary Fig. 20). Furthermore, the hydrogen bonding interactions between the side-chain of H95 and the side-chain

of T99 (HB1) were abolished (Fig. 6g, Supplementary Fig. 20b). However, likely due to weaker D222–H95 contact, the voltage-dependent side-chain reorientations of R92 (Supplementary Fig. 18e) and minor water occupancy in the retinal pocket (Supplementary Fig. 24b) was recovered.

Another fascinating mutation pair focusing on the position D95X extracted from Arch5[17] and *Varian3/Varian4*[12] is D95E and T99C (Fig. 6a). In Arch5, this double mutation showed red-shifted absorption spectra and increased fluorescence QY[17]. Likewise, the Archon1-Q95E-T99C displays a red shift of the absorption maxima by 18 nm accompanied by an increase of the fluorescence QY to 0.61% (Fig. 6c). Despite the introduction of the negative charge at the primary counterion position, we did not observe any photocurrents (Supplementary Fig. 30). The construct with improved properties and increased fluorescence with 620 nm excitation (Fig. 6d) was named ARies1 (*AR*chaerhodopsin-3 with *i*mproved *e*xcited-*s*tate). The overall voltage sensitivity of this mutant was decreased to $\Delta F/F_{max} \approx 11.4\%$, whereas the voltage sensitivity in the application relevant range between −60 and 0 mV almost matches that of Archon1. The addition of the promising mutations D125N did not further modify the voltage sensitivity of ARies1, whereas the addition of T100S almost completely abolished voltage sensing of ARies1 (Fig. 6f). These results indicate that both D125 and T100 are the functioning voltage-sensing residues in ARies1 (HB2, HB3, and HB4) (Fig. 6g and Supplementary Fig. 20c).

The MD simulations on ARies1 revealed different behavior of the hydrogen bonding network and a decrease in its voltage sensitivity. The most prominent differences compared to Archon1 were: (a) the interaction between Q95E and T99C (HB1) (Fig. 6g) (b) the breakage of the D222–E95 contact (Supplementary Fig. 20c), and (c) the inward orientation of R92 (Supplementary Fig. 18f). This rearrangement of the hydrogen bonding network allowed water to enter the retinal cavity (c). Furthermore, instead of D222, E95 established a strong hydrogen bond with the RSBH[+] (Fig. 6h).

## Discussion

Our presented experiments and simulations provide substantial insights into the origin of fluorescence and its voltage sensitivity in microbial rhodopsins. The parameters contributing to fluorescence modulation at zero voltage are in line with previous experiments on BR and other microbial rhodopsins, whereas the diversity of the parameters that modulate the fluorescence upon voltage changes is complex and widely unexpected.

We identified the interaction between the RSB(H[+]) and the counterions (D95 and D222) as well as the interaction between the two counterions as the key factors controlling the fluorescence intensity in Arch3 and its variants. The significance of the interactions has been already discussed in the context of ion transport of microbial rhodopsins. D95X, where X stands for arbitrary amino acids introduced at this position (Supplementary Fig. 1), mimics protonation of the dominant RSBH[+] counterion and affects the excited-state lifetime, retinal isomerization pathways, voltage dependency, peak absorption, and the Schiff base p$K_a$ (Fig. 1, Fig. 6, Supplementary Fig. 29)[2]. The elimination of the negative charge of the first counterion (D95) promotes D222 to act as the primary and only counterion, changing the geometry, the hydrogen bonding network, and the water content of the active site.

Consequently, the excited-state of QuasArs, all with D95X mutation, is extraordinarily long-lived with biphasic decay (4–15 and 40–75 ps), which leads to high fluorescence QY and may reach almost 1% (Fig. 1d–f, Supplementary Figs. 4 and 7). During the extended excited-state lifetime, most of the molecules (>95%) return to the original ground state, and only a minor fraction (<5%) undergoes chromophore isomerization. Similarly, the counterion mutants BR-D85S (~14 ps)[39,40] and BR-D85N (~10 ps)[41] also show extended excited-state lifetimes. Additionally, acidified BR$_{605}$ with protonated D85 also

exhibited a prolonged excited-state lifetime of ~20 ps, a ten-fold increase in fluorescence QY, and reduced efficiency of K-state formation[20]. Thus, as confirmed by our experiments, the removal of the primary counterion alone is shifting the absorption maximum to a longer wavelength, prolonging the excited-state lifetime, and increasing fluorescence. Furthermore, D95Q substitution allows to the establishment of a hydrogen bond between D222 and Q95, which completely removes the water from this site.

It has been proposed by QM/MM simulations of the fluorescent Anabaena Sensory Rhodopsin mutant ASR-W76S/Y195F that the modification causes excited singlet states S$_1$ ($^1$B$_u$) and S$_2$ ($^1$A$_g$) coupling, which leads to increased fluorescence[42]. The resulting mixing between the two states temporarily traps the excited molecules in the disturbed S$_1$ state and results in an extended excited-state lifetime[42]. In our experiments, we observed a close correlation between one and two-photon excitation spectra of QuasAr2 (Fig. 1a) in agreement with such a proposition. The close matching of one and two-photon excitation spectra is unusual for protonated retinylidene chromophores[43,44]. A previous study on BR demonstrated that the two-photon excitation spectra are blue-shifted relative to the one-photon spectra[43]. Thus, in QuasAr2, it is likely that there is a similar situation to the one in the ASR double mutant where the singlet excited states S$_1$ ($^1$B$_u$) and S$_2$ ($^1$A$_g$) are coupled. However, we cannot exclude that both one and two-photon excitations promote QuasAr2 to the same excited singlet state.

As explained in the model of Fig. 7a, the illumination may induce single and double isomerization, however, with very low efficiency. Moreover, both the QuasAr IDS as well as the dark state reached after photoequilibration are comprised of a mixture, albeit different, of all-*trans*/15-*anti* and 13-*cis*/15-*syn* retinal isomers, as revealed by retinal extraction and pre-resonance Raman spectra (Fig. 2, Supplementary Fig. 5). The interconversion is based on two double bond isomerization primarily C$_{13}$=C$_{14}$ and C$_{15}$=N. The resulting photoproduct would maintain similar chromophore geometry and charge distribution and would be almost spectroscopically indistinguishable from the IDS. Detailed studies of absorption spectra changes of QuasAr1[23] and Archon1[45] upon illumination support the proposed heterogeneity of the protonated species.

The single isomerization around C$_{13}$=C$_{14}$ or any other double bond would drive a photocycle and result in deprotonation with P400 as the final photoproduct (Fig. 2c). Single isomerization and deprotonation would lead to a mixture of 13-*cis*/15-*anti*, 11-*cis*/15-*anti*, 9-*cis*/15-*anti*, and 13-*trans*/15-*syn*, all with subsequent RSBH[+] deprotonation to RSB. This diversity is attributed to the lack of negative charge at the primary counterion position D95X. Our observation is in line with BR, where the role of the counterion complex has been proven to steer isomerization specificity around the C$_{13}$=C$_{14}$ bond[46]. Furthermore, illumination of acidified BR$_{605}$ resulted in the formation of photoproducts with 11-*cis* and 9-*cis* retinal isomers in addition to the typical 13-*cis*[47].

An even more complex isomerization pathway is observed during photo-recovery of the dark state-like species (D590′, Figs. 1b, c, 2d). After the blue illumination of P400, the single isomerization favors the population of 13-*cis*/15-*syn*. This is caused by C$_{13}$=C$_{14}$ isomerization of 13-*trans*/15-*syn*, or C$_{15}$=N isomerization of 13-*cis*/15-*anti*, as similarly shown in HKR1[48,49]. Most likely 11-*cis*/15-*anti* and 9-*cis*/15-*anti* undergo back isomerization to 13-*trans*/15-*anti* because the 11-*cis* and 9-*cis* isomers disappear and no double-cis isomers were detectable.

To assess the voltage-induced changes in microbial rhodopsin-based fluorescent sensors, multiple parameters have been investigated. The MD simulations revealed that the membrane voltage modulates the rearrangement of residues and the hydrogen bond network in the RSBH[+] proximity with D/Q/H95-T99 as the central linkage. However, we did not experimentally observe a correlation between the increase of voltage-dependent fluorescence intensity and

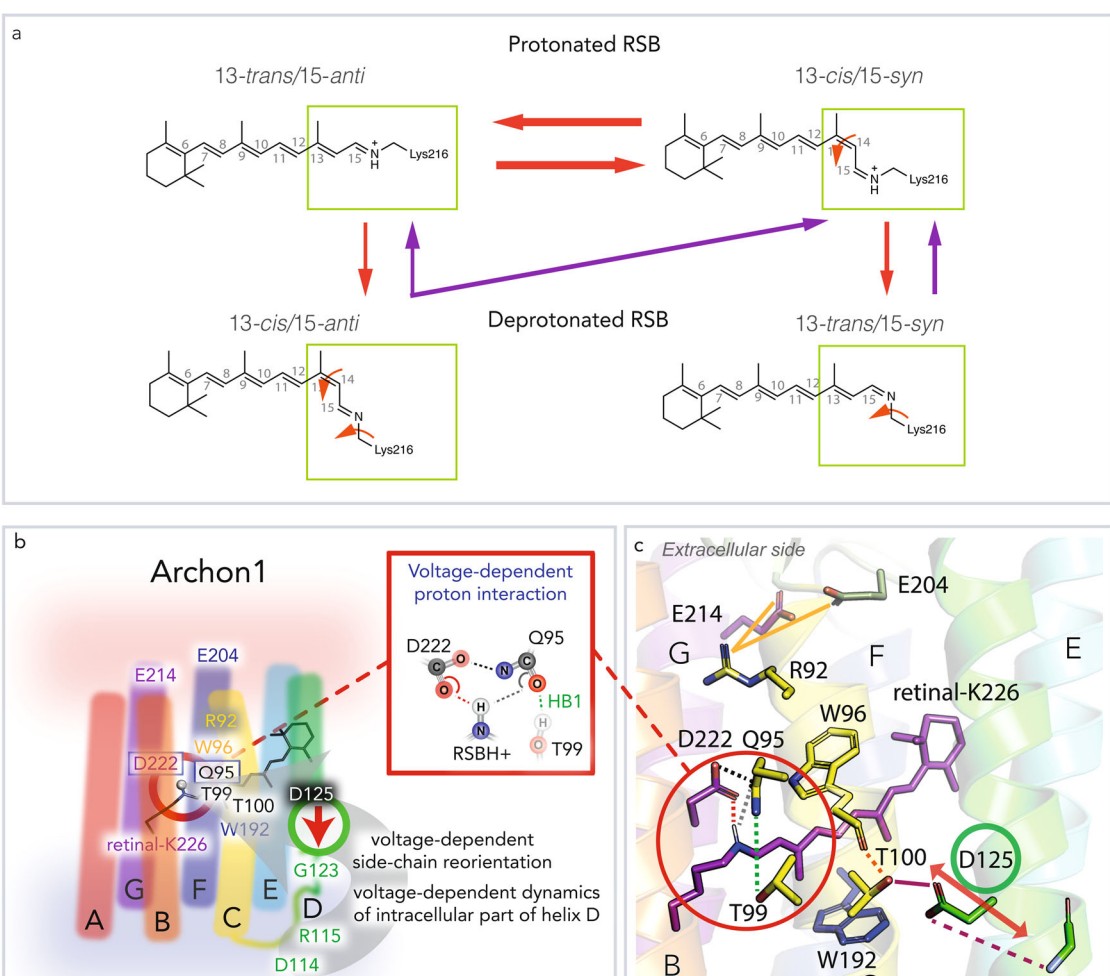

**Fig. 7 | Equilibrium between retinal isomers and dynamics at the retinal binding pocket. a** Proposed scheme of retinal isomer equilibrium formed after illumination. The isomerization pathways taking place after red (625 nm) illumination of protonated species are indicated with red arrows. The isomerization pathways taking place after blue (400 nm) illumination of the deprotonated species are indicated with violet arrows. The thickness of the arrows represents the likelihood of the isomerization to take place, as predicted from HPLC analysis of the extracted chromophore (Fig. 2). Nevertheless, the efficiency of all isomerization pathways is very low. The red illumination may induce double and single isomerization. The more likely double isomerization would result in an equilibrium shift between the 13-*trans*/15-*anti* and 13-*cis*/15-*syn* (upper row of the panel). The single isomerization would drive a photocycle and result in deprotonation (a shift between the upper row indicated isomers and the lower row indicated isomers). The blue illumination of deprotonated species would result in a backreaction with a strong preference towards the population of 13-*cis*/15-*syn* isomer. **b** According to our models, the voltage-dependent dynamics of the intracellular part of helix D modulate the side-chain orientation of D125 in helix D. The reorientations of D125 in Archon1 result in the reorganization of the hydrogen bonding network involving the RSBH[+]. The key residues are indicated. **c** The key residues and their interactions are indicated in the homology model of Archon1.

fluorescence kinetics, retinal isomer composition, or color tuning of the protonated chromophores.

As it was considered for a long time in microbial rhodopsins[50,51], R92 was found to be the main voltage sensing residue in the wt Arch3. According to our simulations, reorientation of the R92 side-chain led to a restructuring of the hydrogen bond network, including voltage-dependent formation and release of the hydrogen bond D95-T99 (HB1) and water intrusion. However, in Archon1, with the D95Q substitution, the R92 side-chain was predicted to remain oriented toward the extracellular space, independent of the membrane voltage. Additionally, in Archon1 counterions Q95 and D222 formed a stable voltage-independent hydrogen bond (Supplementary Fig. 21). In difference to Arch3, the MD simulations predicted a membrane voltage-dependent reorientation of the D125 side-chain (helix D). The flipping of D125 likely results from the increased flexibility of the lower part of the helix D (comprising several charged residues) upon positive voltages (Supplementary Fig. 15d). The conformational changes modulate the hydrogen bonds between T100-D125 (HB3) and T100-W96 (HB2), and most importantly also Q95-T99 (HB1) (Fig. 3a, b, e, Fig. 7b, c). The formation of HB1 at a positive voltage was accompanied by the increased rigidity of the chromophore environment and by a strengthened RSBH[+]-D222 hydrogen bond (Supplementary Fig. 16). The MD simulations combined with experimental Archon1 mutant studies (Figs. 3, 4, 6, Supplementary Fig. 31) allowed us to identify Q95, T99, T100, and D125 as the key residues involved in the voltage sensing (Fig. 7b, c). However, the exact way how they modulate the voltage-dependent fluorescence remains partially open. To investigate this further in-depth investigation combining classical MD and QM/MM methods should be carried out.

Contrary to our anticipation[52], the spectroscopic investigation revealed that the membrane voltage modulated fluorescence intensity does not result in major changes in a mean fluorescent lifetime (Fig. 5, Supplementary Fig. 26), absorption shift (C=C stretch in Supplementary Fig. 28), or retinal isomer composition (C−C stretch in Supplementary Fig. 28). The absence of substantial kinetic changes reveals that membrane voltage does not modulate the excited-state lifetime,

but points towards a voltage-modulated population of fluorescent species. Here we do not consider the possible Stark effect on the excited state and the fluorescence intensity. We think it is unlikely according to previous work by El-Tahawy et al. 2016[53], which reported no effect of vertically applied electric field on the energy levels $S_0$ and $S_1$ of retinal chromophore. However, the possible Stark effect on excited electronic states in QuasArs and Archons could be further investigated by QM/MM studies. Consequently, we considered voltage-dependent fluorescence quenching and voltage-modulated change in RSB protonation.

It is known that tryptophan can form non-fluorescent complexes with organic dyes in the electronic ground state[52,54]. Interestingly, we observed static quenching[55] of all-*trans* retinal fluorescence with increased tryptophan concentration in the Stern−Volmer plot (Supplementary Fig. 26e). The adjacent W96 and W192 could act as possible static fluorescence quenchers, forming a non-fluorescent dark state configuration. However, from the MD simulations we did not observe correlation between the fluorescence properties and a voltage-dependent orientation of W96/W192 (Supplementary Figs. 32 and 33). Furthermore, in highly fluorescent far red absorbing NeoR (QY-20%), the two tryptophans were found to be conserved (W137 and W234) and the fluorescence was not quenched[18].

Thus, it is most likely that voltage regulates the equilibrium between protonated and deprotonated RSB. This should explain our observation that the fluorescence intensity decreases under negative membrane voltage (Fig. 5a, b), while the fluorescence lifetime remains the same (Fig. 5c, d). Negative membrane voltage should promote deprotonation of RSB and result in a decrease in the number of fluorescent QuasAr/Archon molecules.

As predicted by MD simulations in Archon1, the strong RSBH$^+$ −D222 hydrogen bond suggests D222 as an alternative proton acceptor upon formation of P400 Supplementary Figs. 16 and 34). This is in agreement with the previously stated hypothesis on microbial-rhodopsin-based voltage sensing mechanism[8,17,21,56,57]. Furthermore, counterion mutant BR-D85N[58] showed electric field-dependent RSB deprotonation. Layers of oriented membranes deposited on a glass substrate were subjected to an electric field of the order of $10^5$ V/cm, estimated to be similar of ~100 mV membrane voltage. Directional electric field, emulating negative membrane voltage, resulted in a strong blue shift of absorption spectra close to 400 nm, indicating deprotonation of RSB. The RSB deprotonation in darkness might be considered unlikely due to the high p$K_a$-9 of the RSBH$^+$ observed in Archon1 (Fig. 3g) and BR-D85N[58]. However, recent nuclear magnetic resonance (NMR) studies on BR have shown that reversible proton transfer between RSBH$^+$ and the primary acceptor D85 can take place to a substantial degree already in darkness despite the reported p$K_a$-13 of the RSBH$^+$[59]. Thus, we conclude that membrane voltage modulates the proton transfer between retinal Schiff base and its counterion, and this transfer is further tuned by the hydrogen bonding network. This is initiated by the reorientation of R92 in Arch3 and D125 in Archon1.

Finally, a still open question is how the membrane voltage modulates the chromophore isomerization pathway. As outlined above, continuous illumination drives both single and double isomerization, whereas only single isomerization results in deprotonation (Figs. 1, 2, and 7a). Single isomerization results in substantial changes of the chromophore geometry and retinal binding pocket, reducing the p$K_a$ of the RSB[59,60] and permitting its deprotonation. Furthermore, the MD simulations have shown that the formation of the Q95-T99 bond at positive voltage increases the chromophore rigidity (Supplementary Fig. 16). Thus, it is likely that membrane voltage controls the environment of the chromophore and its rate and preference of isomerization. As such, the voltage may modulate the isomerization preference and further fine-tune chromophore deprotonation during extended illumination periods.

## Methods

### DNA preparation for *E. coli* expression

For purification and *E. coli* cell assay, an *E. coli* codon-optimized genes encoding for Arch3 (UniProtKB−P96787 amino acids 1−253, mutations adapted as listed below) with a TEV protease cleavage site and a 6xHis tag at the C-terminal (ENLYFQGLEHHHHHH) were synthesized by (GenScript, Nanjing, China) and cloned into a pET21a(+) expression vector between the NdeI and SalI restriction sites. The constructs were adapted as listed: pET21a (+)-Archon1 (mutations introduced from ref. 12). Further variants of Archon1 were modified using QuickChange site-directed mutagenesis kit (Agilent Technologies, USA).

pBAD-His-B-QuasAr1 (AddGene catalog #64135[2]) and pBAD-His-B-QuasAr1 (AddGene catalog #64134[2]) were a gift from Adam Cohen. The sequences of QuasAr1 and QuasAr2 were sub-cloned in pET21a+ via NdeI/SalI restriction sites. pET28b (+)-Archr3 was a gift from Adam Cohen.

### DNA preparation for ND7/23 expression

For patch-clamp fluorescence recordings, gene pCAG-Archon1-KGC-EGFP-ER2-WPRE was ordered from AddGene (catalog #108423). The Archon1 sequence was afterwards sub-cloned into pEYFP-C1[61,62] plasmid vector in frame with eYFP fluorophore using restriction enzyme free Gibson cloning. Additional sequence fragments were introduced to further improve membrane targeting. A fusion of CrChR2 and CrChR1 sequence fragment (C2C1)[61–63] and a Golgi export signal (GE)[64] was introduced to the N-terminus. Additionally, ER[61,62,65] and TS[61,62,64] trafficking signals were added to the C-terminus. The final construct used for measurements in ND7/23 cells was C2C1-GE-Archon1-TS-eYFP-ER.

### Site-directed mutagenesis

For site-directed mutations, we used QuikChange site-directed mutagenesis kit (Agilent Technologies, USA) according to the manufacturer's instructions. The designed forward and reverse primers, to introduce the desired point mutations, were synthesized by Integrated DNA Technologies (Iowa, USA).

### Protein purification

For protein purification, the *E. coli* codon optimized constructs were chosen for high efficiency and yield. The expression plasmid (pET21a (+)) with the chosen fluorescent variant (QuasAr1/QuasAr2/Archon1/ Arch3) was transformed into C41 (DE3) *E. coli* cells (Luicigen Over-Express C41 (DE3) Chemically Competent Cells (Catalog Number #60442-1-LU), vendor−BioCat GmbH, Heidelberg, Germany). To induce protein expression, LB media was supplemented with 0.5 mM isopropyl β-D-thiogalactopyranoside (IPTG; Carl Roth GmbH, Karlsruhe, Germany) and 5 μM all-trans retinal (ATR; Sigma-Aldrich, St. Louis, USA). This step was followed by 4 h of expression at 37 °C. The harvested cells were then disrupted using an EmulsiFlex-C3 Homogenizer (AVESTIN Inc., Ottawa, Canada). To collect the membrane fraction, we used ultracentrifugation (45,000 rpm/235,000×$g$) for 1 h at 4 °C (Type 45 Ti; Beckman Inc., IN, USA). The protein was solubilized with 1.5% n-dodecyl-β-D-maltoside (DDM, GLYCON Biochemicals GmbH, Luckenwalde, Germany), and 0.3% cholesteryl hemisuccinate (CHS, Sigma-Aldrich, St. Louis, USA). The protein was purified using an ÄKTAxpress protein purification system (GE Healthcare Life Science, Chicago, USA) equipped with a HisTrap HP Ni-NTA column and a HiPrep 26/10 Desalting column. The purified protein was collected in buffer consisting of 50 mM Tris−HCl (pH 8.0), 150 mM NaCl, 0.02% DDM, 0.004% CHS, and 0.1 mM PMSF.

2,5% (w/v) SMA (2:1) copolymer was used to solubilize the protein in SMA nanodiscs. The wet membrane concentration was adjusted to 10 mg/mL. The final buffer consisted of 50 mM Tris−HCl (pH 8.0), 150 mM NaCl. SMA (2:1) was a gift from Polyscope Polymers B.V. (Geleen, Netherlands).

QuasAr1 and QuasAr2 solubilized in DDM&CHS micelles were used for steady-state UV–Vis and for HPLC studies. QuasAr1 and QuasAr2 solubilized in SMA nanodiscs were used for fs-pump-probe and pre-resonance Raman spectroscopic studies. Archon1 and its variants were studied after DDM&CHS micelle solubilization.

Similar procedure was followed for pET28b (+)–Arch3, except the selected *E. coli* strain, was BL21 (DE3) and expression was prolonged to 5 h.

To change the buffer conditions, a rapid buffer exchange was performed using PD-10 Desalting Columns with Sephadex G-25 resin (GE Healthcare Life Science, Chicago, USA).

### *E. coli* cell assay sample preparation

The *E. coli* codon-optimized pET21a (+)-Archon1 was transformed into a C41 (DE3) cell strain. The protein expression was induced with LB media was supplemented with 0.5 mM IPTG and 5 µM all-*trans*-retinal. The expression was done overnight at 28 °C. To ensure freshly grown cells were used for the measurements, the *E. coli* cells were harvested with centrifugation for 10 min with $6000 \times g$ at 22 °C. The cell pellet was resuspended in buffer consisting of 50 mM Tris–HCl (pH 8.0), 75 mM NaCl, 75 mM KCl.

### Mammalian cell culture

The ND7/23 is a mouse neuroblastoma and Rat neuron hybrid cell line (catalog number #92090903, Sigma-Aldrich, St. Louis, MO, USA). It was cultured in Dulbecco's Modified Eagle Medium (DMEM) supplemented with 5% (v/v) fetal bovine serum (FBS) and 1 µg/ml penicillin/streptomycin at 37 °C under a 5% $CO_2$ atmosphere. For patch-clamp fluorescence recordings, $0.5 \times 10^5$ cells/ml were seeded on poly-D-lysine-coated glass coverslips in 35-mm Petri dishes. The next day, the cells were transfected using the FuGENE® HD Transfection Reagent (Promega, Madison, WI, USA; 6 µl FuGENE/2 µg DNA per dish) with the human codon adapted C2C1-GE-Archon1-TS-eYFP-ER (or a derivative mutant). The cell medium was supplemented with 2.5 µM *all-trans-retinal*. The measurements were performed ~48 h later.

### Proteoliposom preparation with Archon1 mutants

Detergent-mediated reconstitution of Archon1, Archon1-D125N, and Archon1-T100S solubilized in 0.02% DDM/0.004% CHS in large unilamellar vesicles (LUVs) with a diameter of about 100 nm was performed according to established procedures for membrane proteins[66]. LUVs were performed with 1,2-dimyristoyl-sn-glycero-3-phosphocholine (DMPC, Avanti Polar lipids, USA) and 1,2-distearoyl-sn-glycero-3-phosphoethanolamine-N-[biotinyl(polyethylene glycol)−2000] (DSPE-Biotin, Avanti Polar lipids, USA). DSPE-Biotin was used for anchoring the proteoliposomes at the surface of the biotin–neutravidin-functionalized FLIM glass slide; for the latter, we adapted a protocol by Götz et al.[67]. For vesicle formation after the thin film method, 1 mg lipids were prepared in 1 ml chloroform at a concentration ratio of 1:1000 (DSPE-Biotin:DMPC) using established procedures to form multilamellar vesicles (MLVs)[68]. To the preformed MLVs in 5 mM Tris pH 7.5, 10 mM KCl, 140 mM tetraethylammonium chloride at a concentration of 1.5 mM lipid, 160 nM solubilized Archon1 proteins were added directly before the FLIM measurements to yield the desired final protein-to-LUV ratio of 5–10 Archons/vesicle. Detergents were removed by hydrophobic adsorption to Bio-Beads (Bio-Beads SM-2 Resin, Bio-Rad, USA) for 30 min [1]. LUV-proteoliposomes were then formed by extrusion through a polycarbonate membrane with a pore size of 100 nm (Avanti Mini-Extruder, Avanti Polar Lipids, USA)[68]. The resulting Archon1-proteoliposomes were analyzed for protein orientation using a covalently bound pH indicator dye at the cytoplasmic surface (FITC bound to only accessible lysines in position 47, 112, or 171), showing a right-side-out orientation (cytoplasmic side in, extracellular side out).

Proteoliposomes were then linked to the FLIM cover slides and washed three times with 5 mM Tris pH 7.5, 10 mM KCl, 140 mM tetraethylammonium chloride. Then valinomycin was added to give a final concentration of 10 nM. To achieve the different membrane voltages, the external buffer solution was exchanged with the different KCl concentrations (0.5, 10, 150 mM) and the respective tetraethylammonium chloride concentration to keep the overall salt concentration constant. The Nernst equation was used to calculate the membrane voltage $V_{membrane}$

$$V_{membrane} = \frac{RT}{zF} \ln\left(\frac{[K]_{out}}{[K]_{in}}\right) \qquad (1)$$

with $R$ the ideal gas constant, $T$ the temperature, $z$ the valence of the ion, $F$ is the Farraday constant. From the respective potassium concentrations inside $[K]_{in}$ or outside $[K]_{out}$ of the proteoliposome, the membrane potentials of −75, 0, and 68 mV were calculated.

### UV–Vis absorption spectroscopy

The UV–Vis spectroscopic measurements were performed at room temperature (~22 °C). The steady-state absorption spectra were recorded using a Cary 300 spectrophotometer (Varian Inc., Palo Alto, USA) at a spectral resolution of 1.6 nm and a UV2450 (Shimadzu, Kyoto, Japan) with a spectral resolution of 0.5 nm.

To monitor the changes in the absorption spectra after light adaptation, the sample was illuminated with a 625-nm LED (Luxeon Star LEDs, Alberta, Canada) for 30 s– 1 min to accumulate the P580/P400, and a 400-nm LED (Luxeon Star LEDs, Alberta, Canada) for 30 s–1 min on the P580/P400 sample for recovery to the D590' state. Data analysis was performed using Excel 2016 and Origin 9.0 (OriginLab Corporation, MA, USA).

### UV–Vis flash photolysis

The transient absorption spectra were recorded as reported previously[61]. We used a modified LKS.60 flash photolysis system (Applied Photophysics Ltd., Leatherhead, UK) to measure changes in the transient absorption spectra in the nanosecond-to-second time range with multi-wavelength datasets (0.4 nm spectral resolution). To excite the sample, the laser pulse was tuned to 580 nm (QuasAr2) and 530 nm (Arch3) with a tunable optical parametric oscillator (MagicPrism, Opotek Inc., Carlsbad, CA, USA). The OPO unit was pumped with the third harmonic of a Nd:YAG laser (BrilliantB, Quantel, Les Ulis, France). The laser was adjusted to energy of 5 mJ/shot and a pulse duration of 10 ns. A 150 W Xenon lamp (Osram, München, Germany) was used as a monitoring light source to track the changes in the absorption spectra. The transient spectra were recorded using an Andor iStar ICCD camera (DH734; Andor Technology Ltd, Belfast, Ireland) with custom software written in Visual C++. In the case of QuasAr2, 46 different time points between 10 ns and 10 s (5 points per decade, isologarithmically) were recorded and averaged over 4 scans (runs). In the case of Arch3, 70 different time points between 100 ns and 1 s (20 points per decade) were recorded and averaged over 7 scans. The sample was kept in the dark for 60 s before the subsequent recording. The measurements were performed at room temperature (~22 °C).

The primary data analysis was performed using MATLAB R2016b (The MathWorks, Natick, MA, USA) to calculate difference spectra and reconstruct the three-dimensional spectra.

### UV–Vis Femto- to microsecond transient absorption spectroscopy

Transient absorption measurements were performed with a femtosecond-to-microsecond pump-probe setup at room temperature (~22 °C) as reported previously[24,69,70]. The samples were filled in a homemade sample holder that has two 1-mm-thick quartz

windows. The sample thickness was set at 400 μm with an appropriate sample spacer. The sample holder was set on a Lissajous scanner that ensures sample refreshment after each laser shot, with a time interval of 60 s between successive exposures to the laser pulses[24,70,71]. A $CaF_2$ plate on a homemade moving stage was used for supercontinuum white light generation, and a selected wavelength region (360–720 nm) was detected by the photodiode array. The time delay was varied up to 1 μs by the optical delay (from −100 ps to 3.3 ns with the minimum temporal step of 50 fs) and the electronic delay (from 12.5 ns to 1 μs with the minimum temporal step of 12.5 ns). The diameters of the pump and the probe beams at the sample position were ~150 μm and ~50, respectively. The central wavelength and the power of the pump beam were set at 610, or 620 nm depending on the sample. The excitation pulse energy was set at ~100 nJ. The instrument response function was ~80 fs, as estimated from global analysis. The spectra were averaged over 4 scans for Arch3 and 10 scans for QuasAr1, QuasAr2, and Archon2. The data collection was done using custom software written in LabVIEW 2015 (version 15.0, 64-bit).

## Global analysis methodology

Global analysis was performed for the femto- to microsecond transient absorption spectra using the Glotaran 1.5.1, Snellenburg et al. 2012 program[72,73]. With global analysis, all wavelengths were analyzed simultaneously with a set of common time constants. A kinetic model was applied consisting of sequentially interconverting, evolution-associated difference spectra (EADS) (i.e., $1 \rightarrow 2 \rightarrow 3 \rightarrow …$) in which the arrows indicate successive monoexponential decays of a time constant, which can be regarded as the lifetime of each evolution-associated difference spectra (EADS)[73,74]. The first EADS corresponds to the difference spectrum at time zero. The first EADS evolves into the second EADS with time constant $\tau_1$, which, in turn, evolves into the third EADS with a time constant $\tau_2$, etc. The procedure clearly visualizes the evolution of the intermediate states of the protein[24,70,73,75]. The standard errors in the time constants were <5%.

## Two-photon excitation

A cleaned indented microscope slide topped by a coverslip was used to hold the sample. It was sealed using silicone and a cellulose nitrate polymer. The two-photon spectra were measured using a confocal setup, which has been described previously[76]. For the two-photon excitation range of 1020–1395 nm, an optical parametric oscillator (IR OPO, APE Berlin) driven by a pump laser (Chameleon Ultra II, Coherent Inc., 800 nm, 4 W, 80 MHz) was used. The excitation beam was cleaned up by a long pass filter (FEL900, Thorlabs), and the power was set to 1 mW by a linear variable neutral density filter (NDL-10C-2, Thorlabs). A telescope system widened and collimated the beam to illuminate the whole objective. The beam was fed into an Olympus IX50 microscope body where a dichroic mirror (T860spxrpt, AHF) reflected the beam into the objective (UPlanApo/IR ×60 1.20 W, Olympus). The emission light was separated from the excitation light by the dichroic mirror. Another telescope system widened the emission light to fit the size of the detector. To remove any remaining excitation light, a short pass filter (HC770/SP, AHF) was installed. The signal was detected using an electron multiplying charge-coupled device camera (iXonEM + 897 back-illuminated, Andor Technology). Each measured wavelength was set individually using the IR OPO. The wavelengths were measured three times over the spectral range. Therefore, the wavelengths were set rising from 1020 to 1395 nm for the first measurement, dropping from 1395 to 1020 nm for the second and rising for the third. The range was measured using increments of 5 nm. For each wavelength, videos of 50 frames were recorded at 31.25 Hz (using Solis

11.9999 software from Andor Technology, Belfast, UK and μ-Manager software). For all videos, an emission area and a background area for background correction were defined. Integration of the areas and division by the number of pixels resulted in arbitrary units of intensity for each wavelength. The corrected intensity values for all wavelengths were merged and normalized to receive a normalized two-photon excitation spectrum. The data analysis was carried out using a custom Python 2.7.16 script and Origin 9.0.

## Fluorescence spectroscopy

Fluorescence spectra of the dark and light-adapted states of the QuasArs, Archons, and Archon1 derivatives were recorded with a vis/NIR fluorescence spectrometer Fluoromax (Horiba, Kyoto, Japan) at room temperature (~22 °C), as reported previously[77]. The fluorescence emission data was collected using FluorEssence2.5.2 software from Horiba Instruments Inc., NJ, USA. The samples were diluted to an OD ≲ 0.05 (per cm) at the visible absorption maximum to prevent reabsorption. The fluorescence quantum yield was measured using a laser dye (Oxazine 1, $\Phi_{FL} = 14.1 \pm 0.8\%$[78]) dissolved in ethanol and archaerhodopsin-3 variant Arch5 ($\Phi_{FL} = 0.87\%$[17]) in the buffer as fluorescent standards. The samples were contained in a disposable plastic cuvette (Brand, 759005) with a 10-mm path length both in the excitation and detection directions. The excitation wavelength varied from 530 to 640 nm with a 2-nm excitation slit and a 10-nm detection slit. To obtain the emission spectra, the wavelength step was set at 5 nm, and the acquisition time was set as 0.1 s for each step. A total fluorescence count was summed up in the entire wavelength region for calculation of the fluorescence quantum yield. The fluorescence quantum yield ($\Phi_{FL,QuasAr}$) was calculated with the following formula[78]:

$$\Phi_{FL,QuasAr} = \Phi_{FL,dye} \times f_{dye}(\lambda_{ex})/f_{QuasAr}(\lambda_{ex}) \times F_{QuasAr}/F_{dye} \times n_{water}^2/n_{ethanol}^2 \qquad (2)$$

$f$ and $F$ represent 1−transmittance and the total fluorescence count, respectively. $n_{water}$ and $n_{ethanol}$ represent the refractive index of water and ethanol, respectively. For Archon1 variants, the fluorescence QY is compared at 580 nm excitation against Archon1. Excitation spectra were taken at 400–700 nm (with a 2-nm excitation slit) with detection at 710 nm (with a 5-nm detection slit) with 3-nm increment steps and 0.2-s acquisition for each wavelength.

Steady state fluorescence experiments with Retinal−BSA complex with 0.8 μM all-trans Retinal (all-trans Retinal, USA, Sigma-Aldrich) and 30 μM bovine serum albumin (BSA, Sigma-Aldrich, USA) with varying tryptophan (Trp, Sigma-aldrich, USA) concentration were carried out with a Fluoromax-3 (Horiba Jobin Yvon, Kyoto, Japan) in 0.3 × 0.3 cm quartz cuvettes (High precision cell, Hellma GmbH & Co. KG, Germany). The fluorescence was recorded between 650 and 800 nm at an excitation wavelength of 640 nm with 1 nm wavelength steps and the acquisition time set to 1 s for each step.

Data analysis was performed using Excel 2016 and Origin 9.0.

## Retinal extraction and analyses by high-pressure liquid chromatography (HPLC)

The HPLC experiments were carried out as previously reported[79]. The dark state sample D590 was not exposed to light before the measurement. To investigate the light-adapted species P590/P400, the sample was illuminated for 30 s with red light LED (625 nm). To investigate the recovered dark-state like species D590', the red light illumination was followed by 1 min illumination with blue light LED (400 nm). Chromophore extraction was achieved, as described in the following steps. First, cooled ethanol (−20 °C) was added to purified protein (solubilised in DDM&CHS micelles). Second, within 2 min, the extracted retinal was dissolved in heptane and 7% (w/w) diethyl ether mixture. Third, after another 2 min, the sample was centrifuged for

1 min (2000 rpm/492 × *g*) to separate the phases. All working steps, except sample illumination and addition of ethanol, were performed under dim red light and at room temperature. Separation of the different retinal isomers in the heptane/diethyl ether mixture was performed using high-pressure liquid chromatography (HPLC) device LC-20AD (Shimadzu, Kyoto, Japan) configured with ReproSil 70 Si, 5 mm column (Dr. Maisch, Ammerbuch-Entringen, Germany). The data was collected using LC LabSolutions from Shimadzu, Kyoto, Japan. The contribution of retinal isomers was estimated by peak integration (Excel 2016 and Origin 9.0).

### Raman spectroscopy

The pre-resonance Raman spectra were recorded with a Fourier transform Raman spectrometer RFS100/S (Bruker Optics GmbH & Co. KG−Ettlingen, Germany). The 1064 nm Nd:YAG laser (DPY 301 II0.50 EM, Coherent, CA, USA) was used as an excitation source. The laser power was set to 640 mW. The scattered light from the sample was collected in back-scattering geometry. The spectral resolution reached with the system is 4 cm$^{-1}$.

The measurements of purified protein were carried out as previously described[80]. The sample concentration was adjusted to an optical density of $OD_{\lambda max}$-30 at the absorption peak maximum (580–590). The sample volume used for the measurement was 6 μM. The samples of QuasAr1 and QuasAr2 were measured at −190 °C to prevent thermal conversion. The spectra were accumulated over 2000 scans to ensure high spectral quality. The measurement of purified Archon1 was carried out at room temperature (-22 °C), to ensure comparability to the intact cell measurements. In this case, the spectra were accumulated only over 300 scans (3 sets of 100 scans) to avoid the incidence of artifacts.

To measure Raman spectra directly in intact cells, the following protocol was established. The cells were prepared as aforementioned. The freshly harvested *E. coli* cell concentration was adjusted to 0.5 g/mL. The sample volume used for the measurement was 1 mL and the cell suspension was filled in a quartz cuvette. The intact cells were measured at room temperature (-22 °C). During the measurements, a digital thermometer with a sensor wire immersed in the sample was recording the temperature over time to ensure that no strong increase in the sample temperature takes place. The spectra were accumulated over 10,000 scans to assure good spectral quality. The spectra were measured 20 times with a fresh sample solution for every 500 scans. To measure the Raman spectra of the sample embedded in the depolarized membrane, 1 mM of 1799 protonophore was added to 0.5 g/mL *E. coli* suspension. After 15 min, the measurement was started (according to the observations made from FLIM measurements Fig. 5). Also, the spectra of this sample were accumulated over 10,000 scans (20 sets of 500 scans) to ensure sufficient signal-to-noise.

The primary data analysis, including baseline correction and averaging over scans, was done with the spectrometer software OPUS (Bruker Optics GmbH & Co. KG, Ettlingen, Germany), and further analyses were performed using Origin 9.0.

### Fluorescence lifetime imaging microscopy (FLIM)

The fluorescence lifetime imaging was performed with a custom-built FLIM setup, which combines time-correlated single photon counting (TCSPC) and confocal laser scanning microscopy as described before[38,81]. A pulsed picosecond supercontinuum laser source NKT SuperK Extreme EXU-3 (NKT Photonics, Birkerød, Denmark) with a repetition rate of 19.5 MHz generated a white-light spectrum for selecting the desired excitation wavelengths. The built-in acoustic-optical tunable filter system (AOTF, UV−VIS Select, NKT Photonics, Birkerød, Denmark) allowed selecting a single excitation wavelength, which was set to 640 nm (or to 540 nm when indicated) to obtain the highest voltage response of Archon1. The

excitation laser beam was focused by an objective (×60 water immersion, UPSLAPO60XW, Olympus, Tokyo, Japan) and scanned (DCS-120, Becker & Hickl, Berlin, Germany) over the sample placed on an inverted microscope (IX71, Olympus, Tokyo, Japan). The emission of the fluorescence was filtered by a 665 nm long pass (665 LP ET, Chroma, Vermont, USA) and a Quad-Notch filter at 405/488/532/635 nm (Quad-Notch Filter 405/488/532/635, Semrock, New York, USA) and was detected by a hybrid PMT detector (HPM-100-40, Becker & Hickl, Berlin, Germany). The pulse width of the white light laser at 640 nm was determined to be ~60 ps (FWHM); the instrument response function (IRF) of the FLIM setup was limited by the response time of the hybrid PMT detector with 120 ps FWHM[38]. The collected photons were sorted into 1024 time channels (width 19.5 ps) by TCSPC modules (SPC-160, Becker & Hickl, Berlin, Germany) using the software Single Photon Counter (Version 9.80, Becker & Hickl, Berlin, Germany).

To measure the FLIM of Archon1 directly in living cells the following protocol was established. The cell preparation is described before. The freshly harvested *E. coli* cell concentration was adjusted to 0.05 g/mL. The sample volume used for the measurement was 100 μL. The intact cells were measured at room temperature (~20 °C). FLIM images were recorded in continuous sets of 15–20 × 1 min measurements. To measure the fluorescence change upon membrane depolarization, 10 μM of gramicidin and the 1799 ionophore were added to 0.05 g/mL *E. coli* suspension.

Proteoliposomes with either Archon1, Archon1-D125N, or Archon1-T100S were prepared as described above[66]. The vesicles were linked on biotinylated cover slides using 0.5 mg/ml Neutravidin (Neutravidin, Sigma-Aldrich, St. Louis, USA) prior for 2 min. After washing three times with water, the vesicles solution was applied for 2 min and was washed afterwards three times with the buffer 5 mM Tris pH 7.5, 10 mM KCl, 140 mM tetraethylammonium chloride. To achieve the different membrane potentials, the external buffer was exchanged with varying KCl and tetraethylammonium chloride concentrations to keep an overall salt concentration of 150 mM constant and measured with the previously mentioned settings in FLIM for 5 min each.

For Retinal−BSA complex experiments, 0.8 μM all-trans Retinal and 30 μM bovine serum albumin was prior mixed in PBS buffer pH 7.5, and before measurements, tryptophan was added from 50 mM stock in PBS pH 7.5 to achieve the desired Trp concentrations between 0.5 and 30 mM. Measuring the Retinal−BSA complex with varying tryptophan concentration in FLIM a pulsed diode laser at 405 nm excitation (BDL-405-SMT, Becker & Hickl, Berlin, Germany) with a repetition rate of 19.5 MHz was used. The emission of the fluorescence was filtered by a 435 nm long pass filter (435LP HQ, Chroma, Vermont, USA) and the previously mentioned Quad-Notch filter. Detection was carried out for 5 min for each Trp concentration as previously stated.

The FLIM data were analyzed using self-written software in C++ and Origin 9.0. Fluorescence decay traces of the individual pixels were partitioned into clusters using a multivariate pattern recognition method[82,83]. To analyze the change in the fluorescence intensity upon membrane depolarization, the obtained image photon counts were normalized by the number of image pixels that captured the *E. coli* cells. Pixels with dark counts of up to three counts within the measurement time of one minute were omitted. Fluorescence lifetime was fitted with a self-written routine to the following equations[52,84] after deconvolution of the fluorescent traces and the IRF.

The time-dependent decay profiles *I(t)* were fitted to a multi-exponential model function

$$I(t) = \sum_{i}^{n} \alpha_i e^{-t/\tau_i} \tag{3}$$

where *n* is the total number of decay components and $\tau_i$ and $\alpha_i$ are, respectively, the fluorescence lifetime and amplitude of the *i*th

component. The population-weighted mean fluorescent lifetime $\bar{\tau}_{pop}$ with the fractional amplitude $\beta_i$ being

$$\beta_i = \frac{\alpha_i \tau_i}{\sum_i^n \alpha_i \tau_i} \qquad (4)$$

was calculated by

$$\bar{\tau}_{pop} = \sum_i^n \beta_i \tau_i \qquad (5)$$

## Fluorescence recordings using streak camera

Time-resolved fluorescence (TRF) measurements at RT were performed with a Hamamatsu C5680 synchroscan streak camera, combined with a Chromex 250IS spectrograph. A grating of 50 grooves per mm and a blazed wavelength of 600 nm was used. The detection range was set from 565 to 834 nm. Measurements were done at two different time ranges: (i) TR1 from 0 to 140 ps (temporal response of 3-4 ps) and (ii) and TR4 from 0 to 1.5 ns (temporal response of 14 ps). The time resolution of the measurements is defined by the temporal responses of the streak camera detection. Femtosecond pump pulses were generated by a Ti:Sapphire laser system, which has been described previously[85]. In brief, mode-locked 800 nm-seed pulses (Coherent-MIRA seed) were amplified via regenerative amplification (Coherent-Rega 9050) and compressed to pulses with a width of ~80 fs. The pump wavelength was tuned to 625 nm by means of optical parametric amplification (Coherent OPA 9400), and the final pump pulses were narrowed around the central value with an FWHM of 10 nm by means of an interference filter (THORLABS). The 625 nm excitation light was vertically polarized, and, by using a lens in front of the sample, the beam was focused into the sample to a spot with typical a diameter of ~100 μm. The laser repetition rate was set at 40 kHz with a laser power of ~3 mW. The fluorescence was collected at 90° from the excitation direction after passing a 640 nm long-pass filter. The data collection was done using HPDTA 9.0 software from Hamamatsu, Hamamatsu City, Japan. 500 μL samples of freshly cultivated Archon1 cells with varying cell concentrations (0.05 g/mL (time base of 1.5 ns)– 0.1 g/mL (time base of 120 ps)) were measured at RT. The sample in the setup was magnetically stirred in a 1 cm × 1 cm cuvette at the speed of 750 rpm, and the excitation laser was focused on the sample close to the cuvette walls. The images (30 s integration of fluorescence per image) were averaged and corrected for background and shading and then sliced into traces of ~2 nm width.

The TRF datasets were globally analyzed with the Glotaran 1.5.1[72]. The methodology of global analysis is described in ref. 73. Additional data analysis was carried out using Origin 9.0. The datasets were analyzed with a parallel exponential model, yielding decay-associated spectra (DAS). For the streak camera results, the instrument response function (IRF) was fitted with a single Gaussian. Average decay lifetimes $\tau_{avg}$ could be calculated from the sum of all lifetimes weighed by the relative amplitudes of each of the DASs: $\tau_{avg} = \frac{\sum (\tau_n \cdot A_n)}{\sum A_n}$ (Eq. (6)).

## System setup for molecular dynamics simulations

Homology models of Arch3, Archon1, and Archon1's variants based on the crystal structure of archaerhodopsin-2 (PDB entry 3WQJ; 1.8 Å-resolution)[34] were built. The chromophore all-trans retinal bound to Lys226 in the crystal structure was retained in all of the models. The homology building using Modeler 9.22[86] was allowed since Arch3 and Arch2 share a sequence identity of 86% and Archon1 and Arch2 share a sequence identity of 80%. In parallel, we predicted the structure of Archon1 using CoLab AlphaFold2[26,27] and RoseTTAFold[28], while the orientation of the retinal was inserted by Modeler 9.22[86].

For each simulation, the homology model was inserted into a pre-equilibrated simulation box of a 1-palmitoyl-2-oleoyl-sn-glycero-3-phosphocholine (POPC) bilayer, water, $K^+$, and $Cl^-$ using the Gromacs tool g_membed[87]. The equilibration of the single membrane system was done as described in the molecular dynamics section.

Afterward, a double membrane system was built by duplicating the equilibrated single membrane system along the membrane normal. Following the computational electrophysiology approach, the transmembrane voltage (TMV) was generated by an explicit ion gradient across the plasma membrane. Here, ion gradients ranging from $q = 0e^-$ to $q = 6e^-$ with a step size of $2e^-$ were added between the two separated compartments. The resulting voltages were in the range of ca. −900 mV to ca. +900 mV. The high potentials were chosen to accelerate the voltage-dependent dynamics of the proteins in a reasonable simulation time. Besides generating a TMV, this setup allows to simultaneously investigate two proteins under different TMVs, −V (negative voltage) and +V (positive voltage).

Additionally to the ion gradient, both compartments contained ions to neutralize the membrane and protein charges and to mimic an ionic strength of ca. 270 mM. Water was modeled with the SPC/E potential[88].

## Molecular dynamics simulations

All classical all-atom MD simulations were carried out with Gromacs 5.1.2 and 5.1.4[89] using the amber99sb force field[90] with the addition of the parameters of all-trans retinal bound lysine[91,92]. Ion parameters[93] and lipid parameters[94] were applied for all of the simulations.

First, a periodic single membrane system (without TMV) was energy minimized in 50,000 steps with a maximum force threshold of 1000 kJ/mol/nm. Then this system was heated to 303 K and thermally equilibrated for 10 ns in an NVT ensemble keeping the number of particles (N), the volume (V), and the temperature (T) constant. During this preparation, position restraints on all heavy atoms with a force constant of 1000 kJ/mol/nm were applied. The subsequent equilibration in an NPT ensemble (keeping the number of particles (N), the pressure (P), and the temperature (T) constant) was conducted for 10 ns with position restraints on all backbone heavy atoms with a force constant of 1000 kJ/mol/nm. The position restraints were sequentially released in another NPT equilibration of 20 ns.

Second, the equilibrated single plasma membrane system was duplicated and piled up along the z-axis, normal to the membrane surface (Supplementary Table 5). The double plasma membrane system was then equilibrated in an NPT ensemble for 20 ns and subjected to a production run of 200 ns in an NPT ensemble. The equilibration and the production run were repeated six times for each charge imbalance to get better statistics. To simulate possible protonation states of neutral histidine residues in Archon Q95H, the equilibration and the production run of δ-nitrogen protonation (HID) and ε-nitrogen protonation (HIE) were repeated three times for each charge imbalance.

Short-range electrostatics and van der Waals interactions were truncated beyond 10 Å. Long-range electrostatics were calculated with the Particle mesh Ewald summation[95,96]. In order to use a time step of 2 fs, all bonds to hydrogen atoms were constrained with the LINCS algorithm[97].

## Molecular dynamics result analysis

The p$K_a$ calculations from the MD trajectories were performed using MDAnalysis[98] and PropKa 3.1[99]. Hydrogen bond interactions were calculated using MDAnalysis with a distance cut-off of 3 Å for donor-acceptor distances and with an angle cut-off of 150° for donor-hydrogen-acceptor angles. Redundant hydrogen bond interactions between a hydrogen donor and the carboxyl oxygen atoms in the same aspartate/glutamate are counted as one instead of two. Inward and outward side-chain reorientations of R92 (or K92 in Archon1 R92K) were determined based on the minimal distances between Cζ of R92 (Nζ of K92) and the extracellular two glutamates (Cδ of E204 and E214)

and the counter ions of RSBH⁺ (Cγ of D95 and Cδ of D222 in Arch3; Cδ of Q95 and Cγ of D222 in Archon1, Archon1-R92K, -T100A, -T100S, -D125N; Cγ of H95 and Cγ of D222 in Archon1-Q95H; Cδ of Q95 and Cδ of E222 in Archon1-Q95E T99C). To collect equilibrated structures, the last 100-ns of individual MD trajectories were only considered for further analysis. Protein structures were visualized with PyMOL 2.1. Data analysis and visualization were performed using matplotlib, numpy, pandas, and seaborn[100–103].

**Whole-cell patch-clamp fluorescence recordings in ND7/23 cells**
Patch pipettes were prepared from standard wall (inner diameter 0.86 nm, outer diameter 1.50 mm) borosilicate capillaries with filament (Science Products GmbH, Hofheim, Germany) using a horizontal P1000 micropipette puller (Sutter Instrument, Novato, USA)[61]. The resistances of the pulled patch pipettes were between 1.5 and 3 MΩ. The reference electrode consisted of an agar bridge with 140 mM [NaCl]. During recordings, membrane resistance was generally >500 MΩ and access resistance was generally <10 MΩ. The recordings were performed in the whole-cell configuration on mammalian ND7/23 cells at room temperature (-22 °C), using an AxoPatch 200B as an amplifier and an Axon Digidata 1440A as a digitizer (both from Molecular Devices, Sunnyvale, CA). A 150 W Xenon lamp (L.O.T.-Oriel GmbH Co. KG, Darmstadt) was used to excite the fluorescent constructs (-0.26–0.28 mW/mm² over all applied $\lambda_{ex}$). The selected excitation wavelength was obtained by choosing an appropriate band-pass filter (FB540-10–FB620-10, FWHM 10 nm, Thorlabs, New Jersey, USA). The excitation light was coupled to the optical path of an inverted microscope Olympus IX70 (Tokyo, Japan) equipped with an Olympus ×40 water immersion objective (Tokyo, Japan). Images were acquired using a pco.panda 4.2 sCMOS camera (Kelheim, Germany). The exposure time was set to 1.5 s per image. To ensure elimination of spectral bleed-through during fluorescence excitation light, a 650 nm long-pass dichroic mirror (DMLP650R and FELH0650, Thorlabs, New Jersey, USA) was used. The intracellular buffer (290 mOsm) contained 110 mM [NaCl], 1 mM [KCl], 1 mM [CsCl], 2 mM [MgCl₂], 2 mM [CaCl₂], 10 mM [HEPES], and 10 mM [EDTA], which was adjusted to pH 7.2. The extracellular buffer (320 mOsm) was set to 140 mM [NaCl], 1 mM [KCl], 1 mM [CsCl], 2 mM [MgCl₂], 2 mM [CaCl₂], and 10 mM [HEPES], adjusted to pH 7.2. The osmolarity of the buffers was modified by using glucose and pH was adjusted by using either citric acid or *N*-methyl-D-glucamine. For recordings, a protocol illuminating the patched cell during image acquisition was started at each voltage step (−80 to +40 mV in 20 mV steps, 10 images per voltage step) with a final repeated −80 mV step to account for bleaching. The data acquisition was done using μ-Manager and pClamp v10.4 Molecular Devices, San Jose, CA, USA.

For each of the constructs, the recording was repeated for 3–14 cells (annotated for each measurement separately in the result section). The recordings were analyzed individually and then averaged. The initial analyses of the images were done using ImageJ. During the signal extraction step, we defined the cell membrane as the region of interest (ROI). For further analysis, we used all pixels in the ROI (with correction for background and bleaching), including pixels that did not show voltage-dependent intensity change. The final analysis of the data was performed using Excel 2016 and Origin 9.0.

**Reporting summary**
Further information on research design is available in the Nature Research Reporting Summary linked to this article.

## Data availability
The structures of archaerhodopsin-2 and archaerhodopsin-3 are available in PDB: 3WQJ, 6GUX. Source data are provided with this paper.

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

## Acknowledgements

We thank, S. Augustin, J. Brandhorst, J. Schäfers, M. Meiworm, and C. Schnick for technical assistance. We thank Martin Heck and Joel Kaufmann for their assistance with HPLC measurements. This work was funded by Deutsche Forschungsgemeinschaft (DFG, German

Research Foundation) under Germany´s Excellence Strategy—EXC 2008/1–390540038 "Unifying Systems in Catalysis" (A.S., P.He., S.H., H.S., and T.U.) and SFB1078—221545957, sub-project A2 (U.A.), and SFB1449—431232613, sub-project A04 (U.A.). We thank the European Community—Access to Research Infrastructures action of the Improving Human Potential Program "LaserLab Europe" (RII-CT-2003-506350) for travel grants and financial support to perform ultrafast pump-probe experiments (A.S. and P.He.). Y.H., P.E.K., and J.T.M.K were supported by the Netherlands Organization for Scientific Research (NWO) through a VICI grant to J.T.M.K. R.F. and P.H. are grateful for funding from the Hector Fellow Academy (30000619, Peter Hegemann). P.He. is a Hertie Professor for Biophysics and Neuroscience and is supported by the Hertie Foundation. S.H., H.S., and T.U. thank the North-German Supercomputing Alliance (HLRN) and Adam Lange for providing us with computational time.

## Author contributions

A.S., P.He. designed and directed the project with contributions from all authors. S.H. carried out MD simulations, and the data was interpreted by S.H., H.S., and T.U. A.S. prepared the biological sample, purified the protein to homogeneity, and performed molecular engineering to obtain the mutants studied. A.S., Y.H., P.E.K., and J.T.M.K. designed and carried out the pump-probe experiments. Data were analyzed by A.S. R.F.L. performed and evaluated the voltage-dependent fluorescence recordings and electrophysiology measurements. A.S., J.B., and U.A. carried out FLIM experiments, FLIM and proteoliposome assay experiments were designed and data analyzed by J.B. and UA. M.T., R.C., and J.T.M.K. designed and carried out fluorescence streak camera recording and analyzed the data. R.M. recorded and together with P.J.W. analyzed the two-photon excitation spectra. A.S., F.V.E., and P.Hi. designed and carried out the pre-resonance Raman experiments. Data were analyzed by A.S. A.S. measured and analyzed the UV–Vis spectra, TAS spectra recorded with flash-photolysis and fluorescence emission, and QY. A.S. carried out an HPLC study of the extracted chromophore and analyzed the data. Data was interpreted by all contributing authors. A.S., P.He., and S.H. wrote the manuscript with contributions from all authors.

## Funding

## Competing interests

The authors declare no competing interests.
