## [Peer Review File · Nature Communications]

REVIEWER COMMENTS

Reviewer #1 (Remarks to the Author):

This manuscript reports a study on opsin-based fluorescent voltage indicators. The study was carefully carried out with the combination of a variety of experiments and molecular dynamics simulations. The authors present comprehensive data on the voltage-dependent fluorescence properties and structural changes and showed that voltage-dependent changes of the hydrogen-bonding networks modulate the protonation state of the retinal Schiff base. I enjoyed reading the manuscript and was impressed at a depth of the presented data. The discussion in the manuscript is sound. The manuscript is suitable for publication in Nature Communications. However, the manuscript should be improved by careful reconsideration and appropriate revision of the text as listed below.

1. The authors claimed that the reorientation of D125 is the key motion for the voltage-sensitive intensity change of Archon1 fluorescence. I understand that the reorientation of D125 can modulate the hydrogen bonding network near the retinal Schiff base and can affect the protonation state of the Schiff base. However, it is unclear the mechanism of the reorientation. Is the orientation just due to the applied electric field? Or are there any other structural changes in the protein to promote the reorientation of D125? There are many charged residues in Archon1. Not all charged residues underwent the reorientation of their side chain upon the change of the applied voltage. Therefore, a detailed examination of the reorientation mechanism of D125 is necessary.
2. The authors presented and discussed the data for QuasAr1 and QuasAr2 in Figures 1 and 2. On the other hand, they presented and discussed the data for Archon1 and its mutants in Figures 3-7. It is not clear how the insights obtained for QuasArs and Archon1 are related to each other. It seems to me that the data for QuasArs are unnecessary to draw the conclusions in this study. The authors should show how the insights for QuasArs and Archon1 are related to each other to draw conclusions.
3. typo in Figure S25c; "inophore" should be "ionophore."

Reviewer #2 (Remarks to the Author):

The paper addresses the origin of fluorescence and its voltage sensitivity in microbial rhodopsins. A combination of time-resolved and stationary spectroscopic studies together with MD homology modeling of the voltage-dependent structure changes and functional patch clamping of cells is interpreted by voltage-induced structural changes of key residues that affect hydrogen-bonded

networks, the fluorescence quantum yield, and the voltage sensitivity of the fluorescence. In addition HPLC chromatograms of retinal extracts provide further key information.

The research addresses an important biological question with applications to optogenetics and biological voltage sensing at the membrane level. The manuscript is clearly written and sharply focused on understanding the molecular basis of the fluorescence of microbial rhodopsins and its modulation by the membrane voltage. The paper is a synthesis of a great deal of information acquired by a team of leading experts involving optical spectroscopy, fluorescence measurements, molecular dynamics (MD) simulations, and patch-clamp functional assays of archaerhodopsin-3 (Arch3) and its various constructs. The team of experts has a high level of technical expertise in all the individual disciplines. Figures are mostly clear and informative and contain a great amount of useful information. Very detailed supplemental information is provided with many additional results that support those presented in the main text.

The major findings and conclusions are highly significant with regard to our understanding of microbial rhodopsins as well as applications to optogenetics and voltage sensing. They include the following:

1. The major conclusion is that membrane voltage leads to the reorientation of R92 in Arch3. This voltage-induced reorientation initiates or modulates the proton transfer between the retinal Schiff base and its counterion and is further tuned by the hydrogen bonding network. This is a very significant finding if correct.
2. Increases in fluorescence quantum yields correspond to extraordinarily long-lived excited states leading to high fluorescence QY (~1%). However, the diversity of the parameters that modulate the fluorescence upon voltage changes are complex and widely unexpected.
3. Excited state lifetimes are established in relation to the interaction between the RSB(H⁺) and the counterions (D95 and D222) as key to controlling the fluorescence intensity in Arch3 and its mutants. The negative counterion (D95) affects the geometry, the hydrogen bonding network, and the water content of the retinylidene binding site.
4. The MD simulations of the homology models together with Archon1 mutant studies identify Q95, T99, T100, and D125 as the key residues involved in the voltage sensing. Removal of the primary counterion shifts the absorption maximum to longer wavelength, and increases the excited-state lifetime, and the fluorescence quantum yield.

5. However, the spectroscopy experiments showed that modulation of the fluorescence intensity by the transmembrane voltage does not correlate with major changes in a excited-state lifetime, or presumably the quantum yield. Additional possibilities involving voltage-dependent fluorescence quenching or voltage-modulated change in retinylidene Schiff base protonation are considered.

General Comments:

Throughout the paper the results of MD simulations tend to be stated as if they were experimental facts or observations, rather than a theoretical result or inference. Please mention that the structural results are inferences from MD simulations of a homology model wherever applicable, and that they do not correspond to actual crystal structures. All figure captions should indicate the temperature and pH of the measurements where applicable.

Specific Comments:

Figure 1, part b - Authors may want to clarify that presumably also P400 is produced, which will help the general reader.

Figure 1 and elsewhere - In the figure caption also, please briefly include the illumination conditions, e.g., duration that the light pulse is switched on. Readers should be able to gather the essentials from the captions.

Page 8, bottom - Authors may also want to mention the Stark effect -- excited electronic states are more polar and affected by voltage and/or electrostatics.

Page 9 and following pages - Authors need to make clear that a homology model is used based on the 1.8 Å crystal structure of Archaerhodopsin-2. To my knowledge there are no crystal structure of either Arch 3 or its any of its mutants.

Page 9, Line 211 - please change to something like: "...is based on homology modeling of the crystal structure of Archaerhodopsin-2, which shares 86% sequence identity with Arch3. Our MD simulations identified..."

Page 9, Line 221 - Change to something like: "In contrast to the homology model of Arch3, ..."

Figure 3, part h - Why is the 280 nm peak so much higher in the R92K mutant? Is the sample less pure? Are there other proteins that affect the voltage sensitivity?

Figure 3, caption - Part e - do the authors mean voltage-independent? Part f - please state in the caption that is patch-clamp of ND7/23 cells to help the general reader.

Page 11, Line 271 - Could the author please comment further - is the protein partially bleached? Or is it impure and the voltage sensitivity (or quantum yield) affected by other components?

Page 22, Line 520 - please change to something like: "According to our MD simulations, reorientation..."

Michael Brown

Reviewer #3 (Remarks to the Author):

The authors Silapetere et al. executed a series of spectroscopy analysis of recently developed Archaeorhodopsin voltage sensors. This line of research seeks to better understand the role of individual mutations on key properties of the sensors, such as quantum yield and sensitivity. The experiments are thorough, and follow in a long line of spectroscopy developed in the Hegemann lab. The authors give some experimental results and molecular dynamics explanations of the results that suggest mutations at key residues either support or disallow voltage driven orientations of amino acids, or regulate the hydrogen bonding environment of the retinal chromophore. This knowledge could be of use in future mutagenesis of the Arch voltage sensor.

However, multiple critiques exist. The novelty of the residues identified as governing Arch sensitivity and the importance of the observed modulations to Archon's performance are unclear. Additionally, there are several major improvements the authors could make to enhance the organization, interpretation, and rigor of this manuscript. The authors should provide additional validations for the predictions of the molecular dynamics experiments and perform more thorough characterization and statistical analysis of the Archon1 mutants. We agree that the expansive set of experiments presented in this work is an odyssey, but better organization and validation would go a long way toward making this work merit publication.

Major comments

1. The authors suggest that understanding the mechanism of voltage sensitive fluorescence is an important target, and we agree. However, the experiments and results of this work leave a lot to be desired. The discussion section leaves the mechanistic understanding still as an open question. The authors do a good job of relating changes in fluorescence lifetime to experimental quantum yield, or putative amino acid positions and bonds to sensitivity. However, the overall range of mutants explored by this work is also somewhat limited, as the mutations were primarily in existing sensors. While a mechanistic explanation of how these mutations support their sensors' enhanced quantum yield or sensitivity is interesting, the manuscript does not fulfill the promise of improved sensors by using their knowledge to improve sensor performance. The reasoning for exploring the R92 residue is somewhat an aside. The authors suggest that R92 prevents water influx in the retinal binding pocket, highlighting that the R92K substitution made Archon less voltage sensitive. However, many mutations could potentially decrease the sensitivity of the sensor, and their mechanism for the decreases would not be an important result for neuroscientists.

2. The authors present a very distributed set of data among their experiments, which seems to be the order of their experiments, but not grouped by type of experiments or results. This lack of organization is below expectations for a high impact work. For example, Fig. S4 and 5 (or Fig. S 15-18,20, or Fig. 3f, 4b) all show the same types of data, but it is difficult to make comparisons between the variants. The data in Fig. 5 and Fig. S23 are meant to show similar effects, but the numerical values for each decay component are not given for comparison. If the results were organized along the types of experiments or analysis for all variants, the message for comparing the effect of point mutations would become much clearer. Along the same lines, the authors jump between fractionated membrane, proteoliposomes, and cells throughout their tests, but it was extremely confusing to try to follow their work. A substantial reorganization that groups experiments and data so that "like appears with like" would greatly help the audience (and the reviewers) interpret the major differences between the variants.

3. In the molecular dynamics experiments, the authors used projected structures of Arch2 as the base model. While simulated predictions of structures have improved in the field and the authors note the high sequence homology between the sensor variants, it is unclear if they are definitive and those could qualify the statements (e.g. about the relative orientations of R92, E204, and E214) about amino acid interaction within the protein. The qualitative observations within the authors' simulations depend strongly on the structure of the variants, which are not validated. The authors could improve the interpretability of the results by confirming the structural predictions across multiple structure software such as AlphaFold and other similar algorithms published in the last year.

4. Given the well-established electrophysiology in this manuscript and past work, the authors should also quantify the accuracy of their molecular dynamics simulations with respect to applied membrane

voltage. The author attempted a wide range of membrane voltages in simulation, are the structural conclusions (voltage dependent orientations) observed at physiological voltages from -100 mV to +100 mV?

5. A substantial number of experiments quantifying sensor performance fails to line up with previous publications. The maximum dF/F of parental Archon1 reported in this paper is about 20-25% the reported sensitivity in the Piatkevich 2018 paper, a large deviation. These experiments are already performed in mammalian cell lines, and should not deviate from published data in a major way. While the authors seem to describe qualitative effects, effectively matching the large dF/F would better support the authors' claims that specific mechanisms impact the sensors properties.

6. For all plots of the Archon variants' sensitivity and quantum yield, the authors should provide statistical analysis (p-value, test) to support their claims that these mutations had a significant effect. The sample size, often $n = 4$ per variant, is quite low for sensor characterization (for example, in Fig. 6c,e,f).

7. One critical component of Arch sensor development has been the sensor response kinetics (often measured by the fluorescence response to voltage steps). While there is not an explicit need to execute the patch clamp voltage step experiments, the authors might comment on the relationship between the photocycle kinetics and the sensor voltage response kinetics.

8. Another critical sensor property is the residual photocurrent, and it is unclear if all of the D95X variants are acceptable as voltage sensors. Some D95X variants retain photocurrent (Kralj et al Nature Methods 2012; Flytzanis et al, Nature Comm. 2014)? Photocurrent analysis of the proposed variants should be a component of rhodopsin analysis.

Minor comments:

Page 3, Line 84: "However, in analogy to BR," — Please define bacteriorhodopsin and other abbreviations the first time they are mentioned.

In Fig S3 panels B & C, the peaks labeled as 580-590 nm appear to the right of the 600 nm tick marks. Some plotting errors may have appeared.

Generally, the panels, legends, and axes of many figures (6, S10, S15-S18, S20, S27, S28) are too small to understand. Please proportion the text labels appropriately.

The authors create the Aries acronym, but fail to use it in many subsequent samples with these mutations. Please be consistent throughout the manuscript.

Vesicle has different spellings in Fig. S23

Editors and reviewers comments are in **black**, answers to the comments from the authors are in **blue**

REVIEWER COMMENTS

Reviewer #1 (Remarks to the Author):

This manuscript reports a study on opsin-based fluorescent voltage indicators. The study was carefully carried out with the combination of a variety of experiments and molecular dynamics simulations. The authors present comprehensive data on the voltage-dependent fluorescence properties and structural changes and showed that voltage-dependent changes of the hydrogen bonding networks modulate the protonation state of the retinal Schiff base. I enjoyed reading the manuscript and was impressed at a depth of the presented data. The discussion in the manuscript is sound. The manuscript is suitable for publication in Nature Communications. However, the manuscript should be improved by careful reconsideration and appropriate revision of the text as listed below.

We thank the reviewer #1 for the overall positive judgment of the manuscript and the suggestions for further improvement.

1. The authors claimed that the reorientation of D125 is the key motion for the voltage-sensitive intensity change of Archon1 fluorescence. I understand that the reorientation of D125 can modulate the hydrogen bonding network near the retinal Schiff base and can affect the protonation state of the Schiff base. However, it is unclear the mechanism of the reorientation. Is the orientation just due to the applied electric field? Or are there any other structural changes in the protein to promote the reorientation of D125? There are many charged residues in Archon1. Not all charged residues underwent the reorientation of their side chain upon the change of the applied voltage. Therefore, a detailed examination of the reorientation mechanism of D125 is necessary.

We sincerely thank the reviewer for pointing out this question. During the revision, we calculated the backbone RMSF values and B-factor for the Archon1 under different transmembrane voltages (Figure R1 c,d). Both RMSF and B-factor describe the flexibility of the protein - high RMSF and B-factor indicate high conformational flexibility. It is obvious to see that one of the most voltage-sensitive parts of the RMSF and B-factor in Archon1 is the lower part of helix D, where D125 is also part of it (Figure R1 a). This can also be seen when we superimpose the end snapshots of all simulations at different voltages (Figure R1 b). While all other six transmembrane helices are well aligned, the intracellular part of the helix D shows considerable structural variation, which is mainly caused by the dynamics of the charged residues at the lower part of the helix D under different transmembrane voltages (namely, D114 and R115 shown in Figure R1 b).

The resulting flexibility in the lower part of helix D perturbs the hydrogen bonding network, which results in a flipping of the side-chain of D125 during the simulations. Helix D is stabilized by the internal hydrogen network including the small residues G119 and G123 (Figure 1a). The lack of side-chains makes these residues more flexible, which also affects the stability of the helix D. Interestingly, the substitution of bulky phenylalanine to valine at position 161 from Arch3 to Archon1 is located near G119 and G123 and might provide more space for the flexibility of helix D in Archon 1 compared to Arch3.

We now include this explanation in the main text as following (lines 250-251) “The MD simulations indicate that movement of D125 originates from the overall increased flexibility of the intracellular part of helix D (Fig. S15b-d).”

And lines 597-600: “In difference to Arch3, the MD simulations predicted a membrane voltage-dependent reorientation of the D125 side-chain (helix D). The flipping of D125 likely results from the increased flexibility of the lower part of the helix D (comprising several charged residues) upon positive voltages (Fig. S15d).”

We also include the figure in the supplement Figure S15.

Figure R1. (a) Hydrogen bonding network of Archon 1 - helix D in the starting conformation of the MD simulation. (b) Superimposition of the end snapshots of Archon 1 at different transmembrane voltages. (c) Backbone RMSF of Archon 1 derived from MD simulations at different transmembrane voltages. (d) Mapping of the B-factor derived from MD simulations on the structure of Archon 1 at positive and negative transmembrane voltages.

2. The authors presented and discussed the data for QuasAr1 and QuasAr2 in Figures 1 and 2. On the other hand, they presented and discussed the data for Archon1 and its mutants in Figures 3-7. It is not clear how the insights obtained for QuasArs and Archon1 are related to each other. It seems to me that the data for QuasArs are unnecessary to draw the conclusions in this study. The authors should show how the insights for QuasArs and Archon1 are related to each other to draw conclusions.

We thank the reviewer #1 for this suggestion. Most studies dedicated on developing microbial rhodopsin-based voltage sensors have been based on random mutagenesis on archaerhodopsin-3 (Arch3), where the key constructs are summarized in Table S1. With this study we would like to provide the reader with a broad study of fluorescence and the origin of its voltage sensitivity in microbial-rhodopsin voltage sensors. In the presented study we have chosen constructs QuasAr1, QuasAr2 and Archon1 from the previously reported microbial rhodopsin-based voltage sensors (Fig. S1).

The constructs were chosen according to the question of interest. For photodynamic studies QuasArs were better suited, as they had less mutations introduced. Furthermore, the counterion mutation (D95H/Q), which results in the different properties of QuasAr1 and QuasAr2 was highly desirable to better understand the photodynamics. However, Archon1 maintained the high voltage sensitivity of QuasAr2, but showed increased fluorescence quantum yield, which is highly desirable for voltage sensitivity studies and experimental studies in cells.

Archon1 is based on QuasAr2, and it conserves the key mutations, in particular the one at the counterion position D95Q as indicated Fig. S1. The additional 7-point mutations introduced in Archon1 are located away from the active site, and resulted in 2.7-fold increased fluorescence QY (QuasAr2 (0.18) - Fig.1h and Archon1 (0.49) - Fig.3i) and 1.9-fold increased excited state lifetime (QuasAr2 (4 ps and 40 ps) - Fig.1e and Archon1 (14 ps and 75 ps) - S8a). Despite the difference in the overall excited state lifetime, the main characteristics are conserved: long lived excited state and blocked pathway for retinal isomerization (QuasAr2 4%, Archon1 0.1%). The similar excited state dynamics allow to correlate the observations between QuasArs and Archon1.

To address this comment the lines 205-212 have been rephrased.

Submitted version: “*We selected Archon1 (Fig. S1b) as the model construct owing to its further improved fluorescence intensity, voltage sensitivity, and membrane targeting. The excited-state dynamics of Archon1 (Fig. S8) are similar to the aforementioned QuasAr2, assuring the compatibility of the developed models. In the case of Archon1, the excited-195 state lifetime is even longer (Fig. S8), and it decays with two time components $\tau_1 = 14$ ps and $\tau_2 = 75$ ps.*”

Updated now: “We selected Archon1 as the model construct owing to its further improved fluorescence intensity, voltage sensitivity, and membrane targeting⁹. Archon1 is based on QuasAr2, and conserves the key mutations at the retinal binding pocket, in particular, the counterion mutation D95Q (Fig S1b). The excited-state dynamics of Archon1 (Fig. S7) are similar to the dynamics of QuasAr2 (Fig. 1e), assuring the compatibility of the developed models. In the case of Archon1, the excited-state lifetime is even longer (Fig. S7), and it decays with two time components $\tau_1 = 14$ ps and $\tau_2 = 75$ ps. Furthermore, the isomerization efficiency decreased from ~4% to 0.1%.”

3. typo in Figure S25c; “inophore” should be “ionophore.”

The Fig. S28c (previous S25c) has been corrected accordingly.

Reviewer #2 (Remarks to the Author):

The paper addresses the origin of fluorescence and its voltage sensitivity in microbial rhodopsins. A combination of time-resolved and stationary spectroscopic studies together with MD homology modeling of the voltage-dependent structure changes and functional patch clamping of cells is interpreted by voltage-induced structural changes of key residues that affect

hydrogen-bonded networks, the fluorescence quantum yield, and the voltage sensitivity of the fluorescence. In addition HPLC chromatograms of retinal extracts provide further key information.

The research addresses an important biological question with applications to optogenetics and biological voltage sensing at the membrane level. The manuscript is clearly written and sharply focused on understanding the molecular basis of the fluorescence of microbial rhodopsins and its modulation by the membrane voltage. The paper is a synthesis of a great deal of information acquired by a team of leading experts involving optical spectroscopy, fluorescence measurements, molecular dynamics (MD) simulations, and patch-clamp functional assays of archaerhodopsin-3 (Arch3) and its various constructs. The team of experts has a high level of technical expertise in all the individual disciplines. Figures are mostly clear and informative and contain a great amount of useful information. Very detailed supplemental information is provided with many additional results that support those presented in the main text.

The major findings and conclusions are highly significant with regard to our understanding of microbial rhodopsins as well as applications to optogenetics and voltage sensing. They include the following:

1. The major conclusion is that membrane voltage leads to the reorientation of R92 in Arch3. This voltage-induced reorientation initiates or modulates the proton transfer between the retinal Schiff base and its counterion and is further tuned by the hydrogen bonding network. This is a very significant finding if correct.
2. Increases in fluorescence quantum yields correspond to extraordinarily long-lived excited states leading to high fluorescence QY (~1%). However, the diversity of the parameters that modulate the fluorescence upon voltage changes are complex and widely unexpected.
3. Excited state lifetimes are established in relation to the interaction between the RSB(H⁺) and the counterions (D95 and D222) as key to controlling the fluorescence intensity in Arch3 and its mutants. The negative counterion (D95) affects the geometry, the hydrogen bonding network, and the water content of the retinylidene binding site.
4. The MD simulations of the homology models together with Archon1 mutant studies identify Q95, T99, T100, and D125 as the key residues involved in the voltage sensing. Removal of the primary counterion shifts the absorption maximum to longer wavelength, and increases the excited-state lifetime, and the fluorescence quantum yield.
5. However, the spectroscopy experiments showed that modulation of the fluorescence intensity by the transmembrane voltage does not correlate with major changes in a excited-state lifetime, or presumably the quantum yield. Additional possibilities involving voltage-dependent fluorescence quenching or voltage-modulated change in retinylidene Schiff base protonation are considered.

We thank Dr. Brown (Reviewer #2) for the careful and positive evaluation of the manuscript. We also appreciate the suggestions that allow to further improve the clarity of the manuscript.

General Comments:

Throughout the paper the results of MD simulations tend to be stated as if they were

experimental facts or observations, rather than a theoretical result or inference. Please mention that the structural results are inferences from MD simulations of a homology model wherever applicable, and that they do not correspond to actual crystal structures.

We thank the reviewer for pointing it out and agree with the reviewer on this point. During the revision we specified the structural results that were obtained from MD simulations.

Submitted version: "At negative voltage, R92 pointed inwards towards the RSBH⁺ and strongly interacted with adjacent negatively charged counterions D95 and D222 (Fig. S11a, d)."

Updated now (lines 236-238): " Our MD simulations predicted that at negative voltage R92 is pointing towards the RSBH⁺ and strongly interacted with the adjacent negatively charged counterions D95 and D222 (Fig. S13a, d). "

Submitted version: "In contrast to Arch3, in Archon1, the R92 faced the extracellular space and interacted with E222 E204 and E214, independent of the applied voltage"

Updated now (lines 244-245): "In contrast to the homology model of Arch3, the R92 flipping was not observed in the simulations of Archon1."

Submitted version: "This contact (HB3) was broken at a positive voltage, and D125 flipped towards the intracellular space establishing a hydrogen bond with the backbone NH of G123 located in helix D (Fig. 3b, c, HB4)."

Updated now (lines 253-256): "At positive voltage, the simulations displayed flipping of D125 towards the intracellular space, disrupting the HB3. D125 establishes new hydrogen bond with the backbone NH of adjacent G123 located in helix D (Fig. 3b, e, HB4)."

Submitted version: "(a,b) Voltage-induced conformations of Archon1."

Updated now (line 273): "(a,b) Voltage-induced conformations of Archon1 as predicted by MD simulations."

Submitted version: "(c) Voltage-dependent probability of important hydrogen bonds (designated below)."

Updated now (lines 277-278): "(c) Hydrogen bonding network as observed by MD simulations at positive transmembrane voltage in Archon1-R92K."

Submitted version: "We have learned that the voltage modulated fluorescence intensity correlates with the rearrangement of residues and the hydrogen bond network in the RSB proximity with D/Q/H95-T99 as the central linkage."

Updated now (lines 585-587): "The MD simulations revealed that the membrane voltage modulates the rearrangement of residues and the hydrogen bond network in the RSBH⁺ proximity with D/Q/H95-T99 as the central linkage."

Submitted version: "Reorientation of the R92 side-chain was observed to lead to a restructuring of the hydrogen bond network, including voltage-dependent formation and release of the hydrogen bond D95-T99 (HB1)."

Updated now (lines 591-593): "According to our simulations, reorientation of the R92 side-chain led to a restructuring of the hydrogen bond network, including voltage-dependent formation and release of the hydrogen bond D95 T99 (HB1) and water intrusion."

Submitted version: "However, in Archon1, with D95Q substitution, the R92 side-chain remained oriented toward the extracellular space, independent of the membrane voltage."

Updated now (lines 594-595): "However, in Archon1, with the D95Q substitution, the R92 side-chain was predicted to remain oriented toward the extracellular space, independent of the membrane voltage."

Submitted version: "Instead, membrane voltage-dependent reorientation of D125 side-chain was observed."

Updated now (lines 597-598): "In difference to Arch3, the MD simulations predicted a membrane voltage-dependent reorientation of the D125 side-chain (helix D)."

Submitted version: "However, we did not observe correlation between the fluorescence properties and a voltage-dependent orientation of W96/W192"

Updated now (lines 625-627): " However, from the MD simulations we did not observe correlation between the fluorescence properties and a voltage-dependent orientation of W96/W192 (Fig. S31, Fig.S32)."

All figure captions should indicate the temperature and pH of the measurements where applicable.

The figure legends Fig.1-6 have been adjusted accordingly.

Specific Comments:

Figure 1, part b - Authors may want to clarify that presumably also P400 is produced, which will help the general reader.

The text in line 108 has been adjusted to "...deprotonation and accumulation of UV absorbing photoproduct P400."

Legend of Fig. 1b (line 127) already contains the reference to P400 "...mixture of P580 and P400 photoproduct ...".

Figure 1 and elsewhere - In the figure caption also, please briefly include the illumination conditions, e.g., duration that the light pulse is switched on. Readers should be able to gather the essentials from the captions.

The figure legends Fig.1-6 have been corrected accordingly.

Page 8, bottom - Authors may also want to mention the Stark effect -- excited electronic states are more polar and affected by voltage and/or electrostatics.

We thank the Reviewer #2 for the suggestion to consider also Stark effect which is in deed of great interest.

Neither from the presented experimental studies, nor from the MD simulations we can draw conclusions on possible Stark effect on the energy levels of Archon1. However, there has been a previous study by El-Tahawy et al. 2016, which reported no substantial effect of vertically applied electric field on the energy levels of the retinal chromophore. This was in contrast to a horizontally applied electric field (arising from the surrounding protein binding pocket). Furthermore, we would expect any changes to the excited state to result in consequential changes in the excited state lifetime/ fluorescence lifetime, which we did not observe. However the suggestion is interesting and could be investigated by QM/MM simulation or vibrational Stark spectroscopy.

We would also like to share with the reviewer, that in a recent collaborative study with Thomas Elsässer we investigated isomerization and early kinetics of bacteriorhodopsin after application of combined THz pulses and photoexcitation. However, the results revealed that the THz pulse did not influence neither the kinetics nor isomerization (unpublished work, P.Hegemann).

The following sentences have been added to the discussion (lines 614-619):

“Here we do not consider possible Stark effect on the excited state and the fluorescence intensity. We think it is unlikely according to previous work by El-Tahawy et al. 2016, which reported no effect of vertically applied electric field on the energy levels S_0 and S_1 of retinal chromophore⁵³. However the possible Stark effect on excited electronic states in QuasArs and Archons could be further investigated by QM/MM studies.”

Page 9 and following pages - Authors need to make clear that a homology model is used based on the 1.8 Å crystal structure of Archaerhodopsin-2. To my knowledge there are no crystal structure of either Arch 3 or its any of its mutants.

Yes, the reviewer is correct. The homology models of Archon1 and Arch3 were generated based on the crystal structure of Archaerhodopsin-2. This information was previously included in the Method section. We added now in the main text “The MD simulations were performed on corresponding homology models, based on the high-resolution X-ray structure of Arch2 (sequence identity of 80%) harboring a protonated all-trans retinal.”. Recently a 3D structure of Arch3 was released which is in a good agreement with our homology model (RMSD=0.5 Å). The comparison between our model and the structure is also included in the SI now (Fig. S12).

Page 9, Line 211 - please change to something like: "...is based on homology modeling of the crystal structure of Archaerhodopsin-2, which shares 86% sequence identity with Arch3. Our MD simulations identified..."

We thank the reviewer for this suggestion and modified the text accordingly. We added that all models are based on the high-resolution Arch2 x-ray structure and that meanwhile the 3D structure of Arch3 has been resolved which perfectly validates our model. Furthermore, following the suggestion from reviewer #3 we compared our models with AlphaFold2 and RoseTTAFold predictions. Also here, a very good agreement is observed. The information including figures is now available in the SI (Fig. S8, S12).

Page 9, Line 221 - Change to something like: "In contrast to the homology model of Arch3, ..."

We changed our formulation accordingly as (lines 244-245) “In contrast to the homology model of Arch3, the R92 flipping was not observed in the simulations of Archon1.”

Figure 3, part h - Why is the 280 nm peak so much higher in the R92K mutant? Is the sample less pure? Are there other proteins that affect the voltage sensitivity?

Indeed, comparing Fig. 3 panels g and h one can notice the increased ratio A_{280}/A_{580} as referred to in the text, line 272. The difference is likely due to a decreased retinal binding stability for Archon1-R92K mutant. Although the mutations in the positions of critical residues do effect the overall expression of the protein, making it more likely to have lower purity even after purification steps. However, once comparing the SDS-PAGE gel of Archon1 and

Archon1-R92K we observe a pronounced band from the desired protein (~ 27 kDa), but we do not observe noticeable contribution of other proteins (Figure R2).

We would like to also point out once more that the absorption spectra between the two samples are compared for solubilized and purified sample, whereas voltage sensitivity is determined for Archon1 constructs directly expressed in ND7/23 cells. The voltage sensitivity $\Delta F/F, \%$ expresses the fluorescence intensity increase compared to the baseline fluorescence intensity (here F_{-80mV}). Only the fluorescence of Archon1 and its variants contribute to the detected fluorescence signal (>650 nm), suggesting that other proteins should not affect the determined voltage sensitivity.

Figure R2. SDS-PAGE gel image of Archon1 and Archon1-R92K.

Figure 3, caption - Part e - do the authors mean voltage-independent?

Indeed, in case of Archon1-R92K hydrogen bonding did not show voltage-dependent changes. However, MD simulations explore the voltage dependency of the hydrogen-bonding network and to maintain cohesive figure legends we kept the expression in Fig. 3e 'Probability of key hydrogen bonds in Archon1-R92K (green) compared to Archon1 (black) at positive, zero or negative transmembrane voltage'.

Figure 3, caption Part f - please state in the caption that is patch-clamp of ND7/23 cells to help the general reader.

Fig.3f caption corrected accordingly.

Page 11, Line 271 - Could the author please comment further - is the protein partially bleached? Or is it impure and the voltage sensitivity (or quantum yield) affected by other components?

As described above, we suggest that the increased A₂₈₀/A₅₈₀ ratio is due to decreased retinal binding. And we do not think that this arises from increased impurities as displayed in the SDS-PAGE gel image (Figure R2).

The ratio A₂₈₀/A₅₈₀ should not affect the determined fluorescence quantum yield (QY). As the QY is determined for particular excitation wavelength as follows:

$$\Phi_{FL,QuasAr} = \Phi_{FL,dye} \times f_{dye}(\lambda_{exc})/f_{QuasAr}(\lambda_{exc}) \times F_{QuasAr}/F_{dye} \times n_{water}/n_{ethanol}^2$$

Furthermore, our observations on purified sample should not affect the conclusions on the voltage sensitivity of the construct determined in ND7/23 cells.

Page 22, Line 520 - please change to something like: "According to our MD simulations, reorientation..."

We changed our formulation according to the reviewer's suggestion as (lines 244-245) "In contrast to the homology model of Arch3, the R92 flipping was not observed in the simulations of Archon1."

Michael Brown

Reviewer #3 (Remarks to the Author):

The authors Silapetere et al. executed a series of spectroscopy analysis of recently developed Archaeorhodopsin voltage sensors. This line of research seeks to better understand the role of individual mutations on key properties of the sensors, such as quantum yield and sensitivity. The experiments are thorough, and follow in a long line of spectroscopy developed in the Hegemann lab. The authors give some experimental results and molecular dynamics explanations of the results that suggest mutations at key residues either support or disallow voltage driven orientations of amino acids, or regulate the hydrogen bonding environment of the retinal chromophore. This knowledge could be of use in future mutagenesis of the Arch voltage sensor.

We thank the Reviewer #3 for the positive evaluation.

However, multiple critiques exist. The novelty of the residues identified as governing Arch sensitivity and the importance of the observed modulations to Archon's performance are unclear.

We appreciate the comment of Reviewer #3. Indeed, the constructs chosen for this study are QuasArs, reported by Hochbaum et al. 2014, and Archon1, reported by Piatkevich et al. 2018. The constructs have been proposed based on random mutagenesis, and have been highly important to establish microbial rhodopsins as fluorescent voltage sensors. However, the

previous reports did not discuss the photodynamics of these constructs or the role of the introduced mutations. With this study we aim to better understand the molecular dynamics leading to the voltage sensitive fluorescence. Furthermore, we want to identify the role of the introduced mutations and key residues, such as the discussion of the counterion D95X mutation. We believe that this study has also led to identify novel key residues, such as T100-D125 pair involved in the voltage sensing. We believe that knowledge on critical residues is crucial for future development of archaerhodopsin-3 based fluorescent voltage sensors.

Additionally, there are several major improvements the authors could make to enhance the organization, interpretation, and rigor of this manuscript. The authors should provide additional validations for the predictions of the molecular dynamics experiments and perform more thorough characterization and statistical analysis of the Archon1 mutants. We agree that the expansive set of experiments presented in this work is an odyssey, but better organization and validation would go a long way toward making this work merit publication.

With this new version, we tried our best to enhance the organization of our manuscript. We included statistical analysis (e.g. Fig. S22, Table S3), validation of prediction of the molecular dynamics (Fig. S8, Fig. S12), prepared new figures, reorganized the old figures and explained our observations more in details in order to increase the readability of the manuscript. We discuss the changes introduced in the points below.

We thank the Reviewer #3 for the valuable suggestions that prompted us to perform a number of new MD simulation analyzes and allowed us to obtain new insights during the revision. In our original submission, we mainly highlighted the voltage-dependent hydrogen bonding network (HB1-HB5) that influences the rigidity of the retinal binding pocket. During the revision, we carefully analyzed the interactions between the Schiff base and the counterions. We found the voltage-dependency of these interactions in Archon1 as well as in most of the other voltage-sensitive Archon1 mutants. These interactions are modulated by the hydrogen bonding network, especially the HB1 (Q95-T99). It should be noted that in rhodopsins, the photon absorbed by the retinal chromophore with protonated Schiff base drives the functional photocycle. The Schiff base proton is stabilized by the negatively charged counterions. Therefore, we think these voltage-dependent interactions between the Schiff base and counterions play a fundamental role in modulating voltage-dependent fluorescence observed in the experiments. Furthermore, following the suggestion from Reviewer #1, we discussed now more carefully about the mechanism of the D125 reorientation in Archon1. The D125 reorientation is caused by the increased flexibility of the lower part of helix D at positive voltages. With all these new findings, we are able to propose a more complete mechanism for voltage-sensitivity in Archon1 and related mutants (Figure R3, main text Fig. 7): Voltage-dependent dynamics of intracellular part of helix D results in side-chain reorientation of D125. The voltage-dependent reorientation of D125 in turn influences the hydrogen bonding network within the retinal binding pocket and the interactions between the Schiff base and the counterions.

Figure R3. Equilibrium between retinal isomers and voltage-sensing mechanism in Archon1.

Major comments

1. The authors suggest that understanding the mechanism of voltage sensitive fluorescence is an important target, and we agree. However, the experiments and results of this work leave a lot to be desired. The discussion section leaves the mechanistic understanding still as an open question. The authors do a good job of relating changes in fluorescence lifetime to experimental quantum yield, or putative amino acid positions and bonds to sensitivity. However, the overall range of mutants explored by this work is also somewhat limited, as the mutations were primarily in existing sensors.

We agree, the constructs chosen for this study have been previously reported (QuasArs, Hochbaum et al. 2014, and Archon1, Piatkevich et al. 2018). The constructs have been proposed based on random mutagenesis, and have been highly important to establish microbial rhodopsins as fluorescent voltage sensors. However, the previous publications there was little evidence and explanation, why the mutation improved fluorescence and its voltage sensitivity, which is in contrast the main point that we want to address in the current manuscript.

With this study we aim to better understand the photodynamics and the molecular dynamics of the voltage sensitive fluorescence. Furthermore, we aim to identify the role of the introduced mutations and the key residues. We provide in depth discussion of the counterion D95X

mutation, and the increased excited state lifetime. From the MD simulations and combination of experimental studies, we could identify several new mutants that affect voltage-sensitivity, such as T100 and D125. This knowledge from combined “computational + experimental” approach we demonstrate here will be highly valuable for future development of next generation archaerhodopsin-3 based fluorescent voltage sensors.

While a mechanistic explanation of how these mutations support their sensors’ enhanced quantum yield or sensitivity is interesting, the manuscript does not fulfill the promise of improved sensors by using their knowledge to improve sensor performance.

This publication was intended to unravel the biophysical principles but not to service the neuroscientists. If the latter is a consequence it would be an add on.

We are convinced that our findings can be potentially used to further improve the sensors for their application.

The reasoning for exploring the R92 residue is somewhat an aside. The authors suggest that R92 prevents water influx in the retinal binding pocket, highlighting that the R92K substitution made Archon less voltage sensitive. However, many mutations could potentially decrease the sensitivity of the sensor, and their mechanism for the decreases would not be an important result for neuroscientists.

Here, we disagree with the Reviewer #3. The R92 position is of general interest for microbial rhodopsins with respect to proton transport and voltage sensitivity.

For example, during the photocycle of BR, upon protonation of retinal Schiff base (RSB) the analogous R82 flips towards the extracellular side triggering proton release from E194&E204 (Clemens et al. 2011 *J. Phy. Chem.*; Hutson et al. 2000 *Biochemistry*). The MD simulations on Arch3 clearly pointed out that this residue changes its orientation upon voltage changes. Such large voltage-dependent reorientations were not observed for other residues, which led us to the prediction that R92 plays a special role in voltage-sensing mechanism. We did not rule out that other amino acids might also play important roles in voltage sensing but we have no evidence for such a statement.

Furthermore, we think that understanding the molecular mechanism of less voltage-sensitive mutants (negative examples) is also highly important, because this knowledge may help us to better understand the system in general, allowing us to propose new mutants with higher fluorescent voltage-sensitivity.

Finally, although the applications refer to neuronal research, the study carried out is basic research, and provides understanding of the key underlying mechanisms of microbial rhodopsins directed to broad scientific community.

2. The authors present a very distributed set of data among their experiments, which seems to be the order of their experiments, but not grouped by type of experiments or results. This lack of organization is below expectations for a high impact work. For example, Fig. S4 and 5 (or Fig. S 15-18,20, or Fig. 3f, 4b) all show the same types of data, but it is difficult to make comparisons between the variants. The data in Fig. 5 and Fig. S23 are meant to show similar effects, but the numerical values for each decay component are not given for comparison. If the results were organized along the types of experiments or analysis for all variants, the message for comparing the effect of point mutations would become much clearer. Along the same lines, the authors jump between fractionated membrane, proteoliposomes, and cells throughout their tests, but it was extremely confusing to try to follow their work. A substantial reorganization that groups experiments and data so that “like appears with like” would greatly help the audience (and the reviewers) interpret the major differences between the variants.

Figures S4 and S5 have been combined as suggested by Reviewer #3.

We thank Reviewer #3 for the suggestion and readily implemented the numerical values for each decay component in Figure 5 c and d. In Figure S26 (old numbering S23) the numerical values of the decay components are now shown in Figure S26 d and for Figure S26 a-c the values are given in the figure legend.

Also the Fig. 3, 4, and 6 have been reorganized to improve the comparability of the results in different variants. For example, the results of the MD simulations of new variants are also shown together with the wt-Archon1 as comparison (Fig. 3, 4, and 6). We also reorganized the supplemental figures (S16-S18, S20, S27, and S28) from the previous version to the new figures (S18-20, S23, and S24) along the types of analysis for all variants, in order to better interpret the major differences between the variants as the reviewer suggested.

3. In the molecular dynamics experiments, the authors used projecte structures of Arch2 as the base model. While simulated predictions of structures have improved in the field and the authors note the high sequence homology between the sensor variants, it is unclear if they are definitive and those could qualify the statements (e.g. about the relative orientations of R92, E204, and E214) about amino acid interaction within the protein. The qualitative observations within the authors' simulations depend strongly on the structure of the variants, which are not validated. The authors could improve the interpretability of the results by confirming the structural predictions across multiple structure software such as AlphaFold and other similar algorithms published in the last year.

We agree with the Reviewer #3 on this point. This project was started at the time point when AlphaFold and RosettaFold were still not (freely) available. We thus use rather a traditional homology modeling approach for generating the unknown structures due to their high sequence similarity (80 %). We followed the reviewer's suggestion and now also performed structural predictions using AlphaFold2 and RoseTTAFold. As shown in the Figure R4 below, which is also now included in the SI as Fig. S8, the predicted structures by AlphaFold2 and RoseTTAFold are highly similar to the structure built by homology modeling with a RMSD of 0.52 Å between homology modeling and AlphaFold2, and a RMSD of 0.60 Å between homology modeling and RoseTTAFold.

Figure R4. Comparison of Archon1 structures generated by homology modeling, AlphaFold2 and RoseTTAFold predictions. (Left) The predicted Archon1 models of CoLab AlphaFold2 (magenta cartoon) and RoseTTAFold (cyan cartoon) are superimposed to the Archon1 homology model (green cartoon). (Right) Comparable orientation and position of the side-chains of the key residues are highlighted in the three models.

4. Given the well-established electrophysiology in this manuscript and past work, the authors should also quantify the accuracy of their molecular dynamics simulations with respect to applied membrane voltage. The author attempted a wide range of membrane voltages in simulation, are the structural conclusions (voltage dependent orientations) observed at physiological voltages from -100 mV to +100 mV?

We thank Reviewer #3 for pointing it out. Within this study we performed a series of MD simulations at different transmembrane voltages ranging from 0 mV and ± 900 mV. The comparison of these simulations at different transmembrane voltages allowed us to identify important residues that are sensitive to the voltage changes. Using the Computational Electrophysiology method, the lowest voltage gradient we could apply is ± 250 mV without drastically increasing the cell size. As discussed in the manuscript, the main reason why we performed the simulations also at very high transmembrane voltage is to accelerate voltage-induced conformational changes in the ns- μ s time scale. We included now in the manuscript that voltage-induced conformational changes were observed in all simulations at different transmembrane voltages (Fig. S11, here in the letter Figure R5), where high transmembrane voltages significantly enhanced the conformational changes. To evaluate whether high transmembrane voltage could influence the overall protein structure, we compared the root mean square deviations (RMSD) of the backbone residues of the simulations performed at different transmembrane voltages as well as without transmembrane voltages. We also overlaid the final snapshots of these simulations. As seen in the Figure R6 (manuscript Fig. S10), there

is no considerable high increase in the RMSD performed at high transmembrane voltages compared to the lower ones or without transmembrane voltage. We conclude that high transmembrane voltage does not have a strong influence in the overall structural integrity, but rather accelerate voltage-sensing process in MD simulations. We also evaluated the stability of the membrane. In our simulation, no electroporation was observed.

We also want to point out that Computational Electrophysiology method has been previously used to simulate ion permeation of a number of K^+ and non-selective cation channels (Köpfer, Science, 2014; Schewe, Cell, 2016; Biedermann, PNAS, 2021). In these studies, systematic comparison was made between the ion permeation simulations performed under different transmembrane voltages. The results showed that high transmembrane voltage did not change the overall ion permeation and voltage-sensing mechanism in these cation channels.

Figure R5. Comparison of the starting structure and the end snapshots of the MD simulations performed at different transmembrane voltages. The end structures of Archon1 (ribbon representation) from the six simulations at different membrane voltages (magenta for negative, cyan for positive) are superimposed to the starting structure (green cartoon representation). The MD simulations were conducted at 303 K by fixing the protonation states of titratable sites at neutral pH.

Figure R6. Comparison of the backbone RMSD derived from MD simulations at different transmembrane voltages for Archon1 and other Archon1 based variants.

5. A substantial number of experiments quantifying sensor performance fails to line up with previous publications. The maximum dF/F of parental Archon1 reported in this paper is about 20-25% the reported sensitivity in the Piatkevich 2018 paper, a large deviation. These experiments are already performed in mammalian cell lines, and should not deviate from published data in a major way. While the authors seem to describe qualitative effects, effectively matching the large dF/F would better support the authors' claims that specific mechanisms impact the sensors properties.

In the voltage change step -80 mV up to $+40$ mV we observe fluorescence intensity change of $\sim 20\%$. However, Piatkevich et al. 2018 reported an increase of the fluorescence in this range of $\sim 50\%$, which is 2-fold higher than we observe (fig.2e). Moreover, the $\Delta F/F_{\max}$ varies between the HEK293T (Fig. 1g, $\sim 80\%$) and hippocampal neuron measurements (Fig. 2e, $\sim 50\%$) within the report, suggesting that cell type and illumination wavelength/intensity can have an effect on Archon1 fluorescence. In this study, we use another cultured cell type entirely: ND7/23 (Mouse neuroblastoma x Rat neuron hybrid cell line). This might have an effect on the voltage sensor properties.

$\Delta F/F$ depends on multiple factors. E.g., the excitation intensity and the sensitivity of camera effect the number of detected photons and consequently S/N. In this report to study the voltage-

sensing properties, we sacrificed illumination intensity for wavelength flexibility. In our case, 150 mW Xenon lamp with wavelength specific band pass filters was chosen as excitation source (0.26-0.28 mW/mm²), which is of much lower power than used by Piatkevich et al. 2018 (for HEK293T: 62 mW/mm², $\lambda_{ex} = 628/31BP$ with an LED; for neurons 637 nm laser 100 mW, 800 mW/mm²).

Also, definition of region of interest (ROI) is critical. In this study the cell membrane (ROI) was selected manually. However, in Piatkevich et al. 2018 a special algorithm was developed to select pixels, which show change in fluorescence intensity. These pixels (ROI) were then chosen for further analysis and determination of voltage sensitivity. However, a general tool user in neuroscience would not use this method.

But this is not the method applied by the general neuroscience user.

Furthermore, we also observed strong dependence of $\Delta F/F$ on the excitation wavelength. Our measurements were carried out with hypsochromic excitation (620 nm vs 628 +/- 31 in HEK cells or 637 nm in neurons), which is likely to result in decreased voltage sensitivity. In addition, Piatkevich et al. consistently use a 664LP for their measurements, while we utilize a 650LP, further deviation from their experimental conditions. In our hands, when using similar wavelength settings as in Fig. 2e of Piatkevich et al. (i.e. 640 nm laser excitation, emission filtering with a 665LP), we achieved a $\Delta F/F_{max}$ of ~50% of recombinant Archon1 reconstituted in proteoliposomes (Fig. S26a), almost identical to reported values in neurons (fig.2e, Piatkevich et al. 2018).

We believe these differences contribute to the deviation observed in the voltage sensitivity. The sensitivity of $\Delta F/F$ on parameters set in different labs, e.g. ROI, has been pointed out already before by Kralj et al. 2011. We believe Archon1 is an exceptional tool with interesting biophysical properties, and therefore hope that our findings enable experimenters to choose their experimental conditions to the advantage of Archon1 constructs, e.g. bathochromic excitation and emission filtering for best results.

6. For all plots of the Archon variants' sensitivity and quantum yield, the authors should provide statistical analysis (p-value, test) to support their claims that these mutations had a significant effect. The sample size, often $n = 4$ per variant, is quite low for sensor characterization (for example, in Fig. 6c,e,f).

We agree with the reviewer that in some instances, the differences are not clear-cut. We have added our statistical analysis for the measured mutations in Figure R7 (supplementary Fig. S22) and Table R1 (supplementary Table S3). Accordingly; we have changed our formulations in the text to better match the statistical analysis.

Figure R7. (a), (c), (e) and (g) maximal $\Delta F/F$ observed for each variant at three different excitation wavelengths (620, 580, 560 and 540 nm respectively). Error is depicted as S.E.M., p-value determination was done via a student's t-test. P-value definitions are as shown in (b). (b), (d), (f) and (h) Cumming estimation plot for the mean difference for comparisons against the shared control Archon1 (A1). Bootstrap sampling distributions were used to plot mean differences. Black dots represent mean differences. The ends of the vertical error bars indicate each 95% confidence interval. Statistical data plotting and analysis was performed using the "Estimation Statistics" online tool [1].

[1] Ho, Jose, et al. "Moving beyond P values: data analysis with estimation graphics." Nature methods 16.7 (2019): 565-566.

Table R1. Statistical data extracted from the estimation statistics analysis using a student's t-test.

620 nm ex.	control_group	test_group	control_N	test_N	effect_size	p-value
1	Archon1	A1 R92K	14	4	mean difference	0.00560237
2	Archon1	A1 Q95H	14	4	mean difference	0.03732736
3	Archon1	A1 T100A	14	4	mean difference	0.13542226
4	Archon1	A1 T100S	14	4	mean difference	0.43684698
5	Archon1	A1 D125N	14	4	mean difference	0.02698261
6	Archon1	A1 QE TC	14	8	mean difference	0.00631080
7	Archon1	A1 QE TC TS	14	3	mean difference	1.63E-05

8	Archon1	A1 QE TC DN	14	4	mean difference	0.02489162
9	Archon1	A1 QE TC DN TS	14	3	mean difference	0.00104760
580 nm ex.						
1	Archon1	A1 T100A	14	4	mean difference	0.83863312
2	Archon1	A1 T100S	14	4	mean difference	0.19934842
3	Archon1	A1 D125N	14	4	mean difference	0.06986549
560 nm ex.						
1	Archon1	A1 T100A	14	4	mean difference	0.71312257
2	Archon1	A1 T100S	14	4	mean difference	0.15903939
3	Archon1	A1 D125N	14	4	mean difference	0.57364073
540 nm ex.						
1	Archon1	A1 D125N	7	4	mean difference	0.02429961

7. One critical component of Arch sensor development has been the sensor response kinetics (often measured by the fluorescence response to voltage steps). While there is not an explicit need to execute the patch clamp voltage step experiments, the authors might comment on the relationship between the photocycle kinetics and the sensor voltage response kinetics.

This is an interesting point suggested by Reviewer #3.

It is important to note that QuasArs and Archons do not undergo a photocycle typical for microbial rhodopsins, as the photoisomerization pathway is “blocked”. Most of the excited molecules return back to the initial dark state, and only a minor fraction undergoes isomerization (in case of Archon1 ~0.05% estimated from fs pump-probe experiments). The excited state lifetime is extraordinary long ($\tau_1=14$ ps and $\tau_2=75$ ps in Archon1 vs 0.3 ps in Arch3). Taking this into account we did not observe a time component that would correlate to the observed kinetic voltage response ($\tau_1=0.61$ ms, $\tau_2=8.1$ ms and $\tau_3=1.1$ ms and $\tau_4=13$ ms in case of Archon1 as reported by Piatkevich et al. 2018).

Although the reported studies are investigating the voltage sensitivity of Archon1 under illumination, we think the voltage-dependent proton oscillation between RSBH⁺ and proton acceptor are light independent.

8. Another critical sensor property is the residual photocurrent, and it is unclear if all of the D95X variants are acceptable as voltage sensors. Some D95X variants retain photocurrent (Kralj et al Nature Methods 2012; Flytzanis et al, Nature Comm. 2014)? Photocurrent analysis of the proposed variants should be a component of rhodopsin analysis.

The D95N discussed by Kralj et al. 2011 is inactive as the authors explained on page 4 and Fig.3 of their manuscript. The lack of the photocurrents and proton pumping is due to the removed primary proton acceptor (D95).

The D95E substitution of the negative charge as described by Flytzanis et al. 2014 is more complex. The construct Archer1 (Arch3-D95E-T99C) showed excitation wavelength dependent photocurrents, where bathochromic excitation (560 nm) resulted in proton pumping, but hypsochromic excitation (655 nm) resulted in voltage sensitive fluorescence. We believe this is due to heterogenous nature of Archer1, which arises from equilibrium of protonated and deprotonated D95E counterion.

In BR-D85E it has been shown that replacement from aspartate to glutamate results in an upshift of the counterion *pKa* value from 3.6 to 6.5 (Subramaniam et al. PNAS 1990). The change in *pKa* results in protonation of the counterion at higher pH values. It may be that the counterion *pKa* is also upshifted in Archer1. The deprotonated D95E can serve as acceptor for

RSB proton and result in proton pumping (photocurrents). However, in case of protonated D95E the construct results in red shifted absorption due to lack of negative charge in RSB vicinity and is no longer able to transport protons.

Taking into consideration the complex nature of D95E mutation we included a supplementary Fig. S30 (Figure R8) to demonstrates the lack of pumping activity in Archon1-Q95E-T99C (ARies1).

Figure R8. A) Exemplary photocurrents for Arch3 and most relevant derivatives. B) Upper panel: Normalized stationary photocurrents at the denoted excitation wavelengths for Arch3 WT, normalized to photocurrents at 550 nm excitation and 0 mV holding potential (N=5). Lower panel: Photocurrent densities for Arch3 WT (N=5), Archon1 (N=3), Archon1 Q95H (N=3) and ARies1 (N=3). C) Photocurrent densities for all variants under all conditions tested.

Minor comments: Page 3, Line 84: “However, in analogy to BR,” — Please define bacteriorhodopsin and other abbreviations the first time they are mentioned.

Line 90 (previous 84) corrected accordingly: “However, in analogy to bacteriorhodopsin (BR) ...”.

Additionally we have included:

Line 84 – “... at the RSBH+ (protonated retinal Schiff base) counterion...”

Line 645– “... recent nuclear magnetic resonance (NMR) studies on BR have shown...”

In Fig S3 panels B & C, the peaks labeled as 580-590 nm appear to the right of the 600 nm tick marks. Some plotting errors may have appeared.

Fig. S3b,c horizontal scale bar has been corrected.

Generally, the panels, legends, and axes of many figures (6, S10, S15-S18, S20, S27, S28) are too small to understand. Please proportion the text labels appropriately.

We resized the text appropriately according to the reviewer’s suggestion.

The authors create the Aries acronym, but fail to use it in many subsequent samples with these mutations. Please be consistent throughout the manuscript.

Line 472 “.. the voltage sensitivity of Archon1-Q95E-T99C..” replace by “.. the voltage sensitivity of ARies1”

Line 472 “.. abolished voltage sensing of Archon1-Q95E-T99C..” replace by “.. abolished voltage sensing of ARies1...”

Line 474 “.. the functioning voltage-sensing residues in Archon1 Q95E-T99C..” replace by “.. the functioning voltage-sensing residues in ARies1...”

Line 487 “... spectra of Archon1 and Archon1-Q95E-T99C with 620 nm excitation...” replace by “... spectra of Archon1 and ARies1 with 620 nm excitation...”

Line 490 “... Archon1-Q95H, and Archon1-Q95E-T99C (ARies1)...” replace by “... Archon1-Q95H, and ARies1...”

Vesicle has different spellings in Fig. S23

Thank you for the hint. We corrected the spelling in Fig. S26b (previous Fig. S23).

REVIEWER COMMENTS

Reviewer #1 (Remarks to the Author):

The manuscript has been well revised. I am of opinion that the manuscript is suitable for the publication in Nature Communications.

Reviewer #2 (Remarks to the Author):

The revision of the paper addresses the major comments and recommendations of the three reviewers and is greatly improved versus the original submission. The authors address the basis of the transmembrane voltage sensitivity of microbial rhodopsins from a fundamental viewpoint, with a focus on archaerhodopsin-3 (Arch3) derivatives and Archon1.

The major conclusion of the work is that the transmembrane voltage modulates the proton transfer between the retinylidene Schiff base and its counterion. This proton transfer is further modulated by the hydrogen bonding network, and it is initiated by the voltage-dependent reorientation of Arginine92 in Arch3 and Aspartate125 in the Archon1 derivative. The insights are important with regards to fundamental understanding of the voltage sensitivity of membrane proteins. From an applications standpoint, the work can guide the development of additional optogenetic tools in neurophysiology.

The critiques of the three reviewers fall into three categories:

1. The conclusion that reorientation of D125 is the motion for the voltage intensity change in fluorescence of Archon1 requires further detailed examination and substantiation.
2. There is a need for experimental validation of the homology modeling used in the molecular dynamics simulations. There is also the need to emphasize that the structural conclusions are based on theoretical MD simulations and not an experimental structure.
3. The overall manuscript organization needs improvement with a stronger conceptual focus, so that it is not just a collection of state-of-the-art spectroscopic and physiological methods applied to an important and highly significant biological problem.

Based on the reviewer comments and the manuscript revisions it would appear that the authors have done all that is asked of them with regards to improving the paper. First, the discussion of the voltage-dependent reorientation of D125 in Archon1 is much improved. It is ascribed to increase flexibility of the lower portion of helix D due to a positive voltage change. Second, the discussion of how the MD simulations lead to structural predictions is also improved. The authors have now emphasized wherever applicable the MD simulation results and have provided two types of validation. Comparison to other simulations involve predicted structures by Alphafold and Rosettafold and are very close to the current structure built by homology modeling. In addition, the recent 3D structure of Arch3 is in very good agreement with the homology model. These results are summarized in the supplemental information of the paper lending greater confidence in the interpretation. Lastly the authors have reorganized the figures in the supplemental information and main text to improve the conceptual organization, with the focus on the biophysical conclusions together with validation of the MD simulations. There is a vast amount of information from diverse techniques and multiple groups, yet the conceptual integration is strong making the work suitable for publication.

Minor comment:

Abstract, Page 1 - Please define "Archon1" in the abstract to avoid confusing the general reader.

Michael Brown

Reviewer #3 (Remarks to the Author):

The authors Silapetre et al present a revision of their manuscript. This revision significantly improved its organization and offered new comparisons and analyses. These additions are valuable, and they have largely addressed my concerns. Some concerns still linger after the rebuttal. Each point starts with quoted text from the manuscript, and follows with reviewer comments/concerns.

1. We included now in the manuscript that voltage-induced conformational changes were observed in all simulations at different transmembrane voltages (Fig. S11, here in the letter Figure R5), where high transmembrane voltages significantly enhanced the conformational changes. To evaluate whether high transmembrane voltage could influence the overall protein structure, we compared the root mean square deviations (RMSD) of the backbone residues of the simulations performed at different transmembrane voltages as well as without transmembrane voltages. We also overlaid the final snapshots of these simulations. As seen in the Figure R6 (manuscript Fig. S10), there is no considerable high increase in the RMSD performed at high transmembrane voltages compared to the lower ones or without transmembrane voltage.

I understand that there are limitations to the simulation. Maybe I missed the point, but what is the reference for the RMSD calculation in Figure R6? Why does the 0 mV simulation follow the same RMSD trace as the other voltages?

2. Moreover, the $\Delta F/F$ max varies between the HEK293T (Fig. 1g, ~80%) and hippocampal neuron measurements (Fig. 2e, ~50%) within the report, suggesting that cell type and illumination wavelength/intensity can have an effect on Archon1 fluorescence. In this study, we use another cultured cell type entirely: ND7/23 (Mouse neuroblastoma x Rat neuron hybrid cell line).

The difference in excitation power is noted, and can indeed be the reason behind the different dF/F signals. A conceptual issue is that if the dynamics change with excitation conditions and power, then the present study is less comprehensive, and would not apply to voltage sensing properties of this rhodopsin probed in neuroscience at much higher powers. The authors should explain this limitation in their discussion.

The choice of the cells should matter less. There is generally a decrease from HEK cell to neurons because neurons tend to express more intracellular aggregates, but the neuron dF/F has been the appropriate number in neuroscience studies. For this study, if the author chose a cell line that has substantial aggregation, manual selection should still mostly bypass these issues. The authors should describe their process in more detail and provide a caveat in the main text that their dF/F calculation was sub-optimal.

3. We have added our statistical analysis for the measured mutations in Figure R7 (supplementary Fig. S22) and Table R1 (supplementary Table S3). Accordingly; we have changed our formulations in the text to better match the statistical analysis.

The authors' choice of t-test probably isn't correct. There isn't any test of normality, nor any expectation that the data would be normal. The low n-values complicate this issue. Please attempt additional non-parametric tests and adjust the statistical conclusions appropriately. Similarly, bootstrapped distributions may not be appropriate in these cases. Finally, there could be multi-comparison corrections that are needed.

Multiple figures (e.g. Fig. 4) are missing comparison bars for matching p-values to comparisons on the graph. Finally, I believe the Nature standard is to show the individual data points for bar graphs where possible, and it is possible for many graphs in the manuscript.

4. The excited state lifetime is extraordinary long ($\tau_1=14$ ps and $\tau_2=75$ ps in Archon1 vs 0.3 ps in

Arch3). Taking this into account we did not observe a time component that would correlate to the observed kinetic voltage response ($\tau_f = 0.61$ ms, $\tau_S = 8.1$ ms ...).

The authors note the difference between the excited state and the voltage response kinetics. However, there are multiple components of the photocycle that are substantially longer than 1 ms in the Arch family, such as the M, N, or O states (Dougal Maclaurin et al, PNA, 110 5939). Could the authors please put the sensor kinetics into context of these other states?

5. Taking into consideration the complex nature of D95E mutation we included a supplementary Fig. S30 (Figure R8) to demonstrate the lack of pumping activity in Archon1-Q95E-T99C (ARies1).

We appreciate the authors' addition of this analysis. However, the comparison to WT Arch is not the most illustrative, as most D95X mutations will substantially decrease photocurrent. However, there are still cases with small residual photocurrent, which is not clearly shown on the new figure. These small photocurrents could still perturb the system in neuroscience studies, for example by activating/inhibiting neurons that are slightly below/above threshold. Please adjust the axes to zoom in on these reduced photocurrents. The n is also very low for these new experiments, but please provide statistical analyses on whether these values are non-zero or not, or a clarification that the presented data do not offer any conclusion on whether the photocurrent is non-zero.

REVIEWER COMMENTS

Editors and reviewers comments in **black**, answers from the authors in **blue**

Reviewer #1 (Remarks to the Author):

The manuscript has been well revised. I am of opinion that the manuscript is suitable for the publication in *Nature Communications*.

We thank the reviewer #1 for the overall positive judgment of the revised manuscript and the recommendation for publication in *Nature Communications*.

Reviewer #2 (Remarks to the Author):

The revision of the paper addresses the major comments and recommendations of the three reviewers and is greatly improved versus the original submission. The authors address the basis of the transmembrane voltage sensitivity of microbial rhodopsins from a fundamental viewpoint, with a focus on archaerhodopsin-3 (Arch3) derivatives and Archon1.

The major conclusion of the work is that the transmembrane voltage modulates the proton transfer between the retinylidene Schiff base and its counterion. This proton transfer is further modulated by the hydrogen bonding network, and it is initiated by the voltage-dependent reorientation of Arginine92 in Arch3 and Aspartate125 in the Archon1 derivative. The insights are important with regards to fundamental understanding of the voltage sensitivity of membrane proteins. From an applications standpoint, the work can guide the development of additional optogenetic tools in neurophysiology.

The critiques of the three reviewers fall into three categories:

1. The conclusion that reorientation of D125 is the motion for the voltage intensity change in fluorescence of Archon1 requires further detailed examination and substantiation.
2. There is a need for experimental validation of the homology modeling used in the molecular dynamics simulations. There is also the need to emphasize that the structural conclusions are based on theoretical MD simulations and not an experimental structure.
3. The overall manuscript organization needs improvement with a stronger conceptual focus, so that it is not just a collection of state-of-the-art spectroscopic and physiological methods applied to an important and highly significant biological problem.

Based on the reviewer comments and the manuscript revisions it would appear that the authors have done all that is asked of them with regards to improving the paper. First, the discussion of the voltage-dependent reorientation of D125 in Archon1 is much improved. It is ascribed to increase flexibility of the lower portion of helix D due to a positive voltage change. Second, the discussion of how the MD simulations lead to structural predictions is also improved. The authors have now emphasized wherever applicable the MD simulation results and have provided two types of validation. Comparison to other simulations involve predicted structures by AlphaFold and RosettaFold and are very close to the current structure built by homology modeling. In addition, the recent 3D structure of Arch3 is in very good agreement with the homology model. These results are summarized in the supplemental information of the paper lending greater confidence in the interpretation. Lastly the authors have reorganized the figures in the supplemental information and main text to improve the conceptual organization, with the focus on the biophysical conclusions together with

validation of the MD simulations. There is a vast amount of information from diverse techniques and multiple groups, yet the conceptual integration is strong making the work suitable for publication.

Minor comment:

Abstract, Page 1 - Please define "Archon1" in the abstract to avoid confusing the general reader.

Michael Brown

We thank the reviewer #2 for the careful, point by point, and positive evaluation of the manuscript and the recommendation for publication in *Nature Communications*. Following reviewer #2 suggestion, we have added a comment regarding Archon1 to the abstract as below.

Lines 31-32 “Molecular dynamics simulations of Arch3, of the Arch3 fluorescent derivative Archon1, and of several its mutants have revealed different voltage-dependent changes of the hydrogen-bonding networks including the protonated retinal Schiff-base and adjacent residues.”

Reviewer #3 (Remarks to the Author):

The authors Silapetere et al present a revision of their manuscript. This revision significantly improved its organization and offered new comparisons and analyses. These additions are valuable, and they have largely addressed my concerns. Some concerns still linger after the rebuttal. Each point starts with quoted text from the manuscript, and follows with reviewer comments/concerns.

We thank reviewer #3 for the careful review and the overall positive evaluation of the revised manuscript. We address the remaining concerns below.

1. We included now in the manuscript that voltage-induced conformational changes were observed in all simulations at different transmembrane voltages (Fig. S11, here in the letter Figure R5), where high transmembrane voltages significantly enhanced the conformational changes. To evaluate whether high transmembrane voltage could influence the overall protein structure, we compared the root mean square deviations (RMSD) of the backbone residues of the simulations performed at different transmembrane voltages as well as without transmembrane voltages. We also overlaid the final snapshots of these simulations. As seen in the Figure R6 (manuscript Fig. S10), there is no considerable high increase in the RMSD performed at high transmembrane voltages compared to the lower ones or without transmembrane voltage.

I understand that there are limitations to the simulation. Maybe I missed the point, but what is the reference for the RMSD calculation in Figure R6? Why does the 0 mV simulation follow the same RMSD trace as the other voltages?

The reviewer is right that the information about the reference structure in previous rebuttal letter (Fig. R6) was missing, but it was present in the main text corresponding figure Fig. S10. The RMSD curves have been calculated taking the equilibrated structures as the

reference. Each MD trajectory undergoes minor structural changes to reach its own equilibrium state of the protein regardless the applied transmembrane voltage. Here we intended to emphasize that the RMSD was calculated for the backbone protein atoms. The important differences caused by the transmembrane voltage affect mostly the side chains of the proteins including the hydrogen bond network. These rearrangements do not or only very slightly affect the overall stability, indicated by the backbone RMSD curve. Therefore, the RMSD traces of the backbone at 0 mV do not differ significantly from the other curves. We added this point to the manuscript (lines 229-236):

“Each MD trajectory undergoes only minor structural changes to reach its own equilibrium state of the protein regardless the applied transmembrane voltage. This is demonstrated by the root mean square deviations (RMSDs) analysis of the protein backbone (Fig. S10) and the comparison of the final snapshots obtained from the simulations performed at different transmembrane voltages (Fig. S11). Here, the proteins behave similar at high voltages and 0 mV. Differences in the protein are, however, visible in the orientation of the side chains and consequently in the hydrogen bonding network as discussed in the following.”

2. Moreover, the $\Delta F/F$ max varies between the HEK293T (Fig. 1g, ~80%) and hippocampal neuron measurements (Fig. 2e, ~50%) within the report, suggesting that cell type and illumination wavelength/intensity can have an effect on Archon1 fluorescence. In this study, we use another cultured cell type entirely: ND7/23 (Mouse neuroblastoma x Rat neuron hybrid cell line).

The difference in excitation power is noted, and can indeed be the reason behind the different dF/F signals. A conceptual issue is that if the dynamics change with excitation conditions and power, then the present study is less comprehensive, and would not apply to voltage sensing properties of this rhodopsin probed in neuroscience at much higher powers. The authors should explain this limitation in their discussion.

The choice of the cells should matter less. There is generally a decrease from HEK cell to neurons because neurons tend to express more intracellular aggregates, but the neuron dF/F has been the appropriate number in neuroscience studies. For this study, if the author chose a cell line that has substantial aggregation, manual selection should still mostly bypass these issues. The authors should describe their process in more detail and provide a caveat in the main text that their dF/F calculation was sub-optimal.

Our aim in the previous correspondence was to explain that multiple parameters contribute collectively to the detected $\Delta F/F$, and the excitation intensity has actually only minor influence on $\Delta F/F$, compared to the wavelength and the region of interest (ROI) for $\Delta F/F$ analysis. Therefore, the reviewer’s #3 concern, “A conceptual issue is that if the dynamics change with excitation conditions and power, then the present study is less comprehensive, and would not apply to voltage sensing properties of this rhodopsin probed in neuroscience at much higher powers”, is not valid. We explain it in detail below.

As suggested previously (Kralj *et al.*, *Nature Methods* 2011), the selection of analysed pixels will result in the largest difference of determined $\Delta F/F$. In our case, the fundamental aspect is that we keep the same data collection protocol for all variants within our study (recording the image under conserved light intensity conditions between excitation wavelengths, primary image analysis in ImageJ, selection of ROI – cell membrane (Fig.R1), correction for background and bleaching and final analysis using Origin 9.0). With this

selection of ROI, we do not rule out pixels that do not respond to voltage change, which might result in lower $\Delta F/F$ values than previous work. Discarding these pixels from analysis could lead to issues for constructs, which shows almost no change in fluorescence intensity upon voltage change.

Additionally, also the excitation wavelength choice results in different $\Delta F/F$ (as shown in the main text Fig. 4c, and previously mentioned in Maclaurin *et al. PNAS* 2013 Fig. 2E). To observe this dependency, we used a light source with tuneable excitation intensity (Xenon lamp with specific bandpass filters for selected wavelengths). And this is an imperative observation to understand the biophysical properties of these constructs. The comparison of the voltage sensor performance under different excitation wavelength conditions allowed important conclusions to be drawn for tool applicants - red light excitation, despite the low absorption, yields higher voltage response (*i.e.*, $\Delta F/F$).

This unusual property for voltage sensors arises due to the heterogeneity of these constructs. The heterogeneity of retinal isomers is discussed in detail for QuasArs, and can be also observed for Archon1 (Fig. S28a, pre-resonance Raman spectra demonstrate mixture of all-*trans* and 13-*cis* retinal isomers). These observations signify, once more, the importance to carry out biophysical studies on the microbial rhodopsin based voltage sensors. An understanding of the mechanism of different constructs will allow in the future a more rational design of the next generation of fluorescent voltage sensors.

On the other hand, excitation light intensity has only a minor contribution to $\Delta F/F$. The fluorescence originates from a single photon process, resulting in linear proportionality between the fluorescence and the excitation intensity. The linear dependence between excitation intensity and fluorescence has been shown for QuasArs (Hochbaum *et al., Nature Methods* 2014 Figure 1b) and Archons (Piatkevich *et al. Nature Chemical Biology* 2018, Supplementary Figure 16h) in previous studies. In such case, both ΔF and F are linearly dependent on the excitation power, thus the excitation power parameter is canceled out in $\Delta F/F$, *i.e.*, intrinsic $\Delta F/F$ is not excitation power dependent. Therefore, the dynamics of sensors do not change with different excitation power, unless extremely high power is used.

When the excitation power is low, the signal-to-noise ratio (S/N) of the measurement decreases, which may influence the accuracy of $\Delta F/F$ measurements, but not to the extent of influencing the dynamics of the sensors. On the other hand, using high excitation intensity is a concern for tool applicants, as it leads to tissue temperature increase, (Owen *et al., Nature Neuroscience* 2019).

Overall, we believe that the selection of region of interest (ROI) and the excitation wavelength are the most critical factors resulting in the difference of the observed $\Delta F/F$ in our study and Piatkevich *et al.*

We add the following comment in the methods section of the manuscript (lines 1172-1175):

“During the signal extraction step, we defined the cell membrane as the region of interest (ROI). For further analysis, we used all pixels in the ROI (with correction for background and bleaching), including pixels that did not show voltage dependent intensity change.”

Fig. R1. Selection of region of interest (ROI) in voltage-clamp experiments on ND7/23 cells.

a) Exemplary fluorescence image of Archon1 at 580 nm excitation with ROI marked in yellow. ROIs were selected manually for each cell, where membrane localization was evident. (As the reviewer #3 mentioned, the neuron cells did show some aggregates, however, the reviewer #3 was also right pointing out that we can bypass it by manual selection.) b) Zoom-in into the selected ROI for acquisitions at different voltages. c) Corresponding $\Delta F/F$ plot extracted from ROI in a) and b) after data analysis.

3. We have added our statistical analysis for the measured mutations in Figure R7 (supplementary Fig. S22) and Table R1 (supplementary Table S3). Accordingly; we have changed our formulations in the text to better match the statistical analysis.

The authors' choice of t-test probably isn't correct. There isn't any test of normality, nor any expectation that the data would be normal. The low n-values complicate this issue. Please attempt additional non-parametric tests and adjust the statistical conclusions appropriately. Similarly, bootstrapped distributions may not be appropriate in these cases. Finally, there could be multi-comparison corrections that are needed.

We understand the Reviewer's concern regarding a test of normality; we have performed one on the datasets at 620 nm, which is where we see a significance in the t-test. According to our testing, we did not see a reason to reject a normal distribution (Table R1).

However, after the suggestion of reviewer#3 and considering the given the datasets, we introduce a nonparametric Wilcoxon rank-sum test (or Wilcoxon–Mann–Whitney test) to substantiate our statistical testing. We have added the results to (here - Table R2, manuscript – Table S3) and used it for our labelling in Figure S22 (here Figure.R2). Finally, we have avoided bootstrapped distributions entirely in our modified figure.

Table R1: Test of normality on 620 nm datasets of all constructs tested in ND7/23 cells with a Shapiro-Wilk normality test.

	N	Statistic	p-value	Decision at level(5%)
Archon1	14	0.95149	0.5841	Can't reject normality
A1 D125N	4	0.91285	0.49763	Can't reject normality
A1 QE TC	8	0.93679	0.57983	Can't reject normality
A1 QE TC TS	5	0.85869	0.22359	Can't reject normality
A1 QE TC DN	4	0.8807	0.34164	Can't reject normality
A1 QE TC DN TS	3	0.8287	0.18504	Can't reject normality
A1 Q95H	4	0.93519	0.62522	Can't reject normality
A1 R92K	4	0.91482	0.50833	Can't reject normality
A1 T100A	4	0.87799	0.33011	Can't reject normality
A1 T100S	4	0.88519	0.36129	Can't reject normality

Table R2: Supplementary Table X: Statistical data extracted from the estimation statistics analysis using a student's t-test and Mann–Whitney U test.

620 nm ex.	control_group	test_group	control_N	test_N	effect_size	p-value (t-test)	p-value (Mann–Whitney U test)
1	Archon1	A1 R92K	14	4	mean difference	0.00560237	0.016872514
2	Archon1	A1 Q95H	14	4	mean difference	0.03732736	0.049451436
3	Archon1	A1 T100A	14	4	mean difference	0.13542226	0.184348097
4	Archon1	A1 T100S	14	4	mean difference	0.43684698	0.366690331
5	Archon1	A1 D125N	14	4	mean difference	0.02698261	0.079725501
6	Archon1	A1 QE TC	14	8	mean difference	0.00631080	0.010483383
7	Archon1	A1 QE TC TS	14	5	mean difference	1.63E-05	0.001402776
8	Archon1	A1 QE TC DN	14	4	mean difference	0.02489162	0.03837024
9	Archon1	A1 QE TC DN TS	14	3	mean difference	0.00104760	0.009801427
580 nm ex.						-	
1	Archon1	A1 T100A	14	4	mean difference	0.83863312	0.873433881
2	Archon1	A1 T100S	14	4	mean difference	0.19934842	0.313027841
3	Archon1	A1 D125N	14	4	mean difference	0.06986549	0.221977464
560 nm ex.						-	
1	Archon1	A1 T100A	12	4	mean difference	0.71312257	0.170418122
2	Archon1	A1 T100S	12	4	mean difference	0.15903939	0.327487682
3	Archon1	A1 D125N	12	4	mean difference	0.57364073	0.647704219
540 nm ex.						-	
1	Archon1	A1 D125N	7	4	mean difference	0.024299608	0.029758067

Fig. R2 Statistical analysis of the recorded voltage sensitivity of Archon1 variants

Statistical testing on datasets obtained on ND7/23 cells. The maximal $\Delta F/F$ observed for each variant at four different excitation wavelengths (620 (A), 580 (B), 560 (C) and 540 (D) nm respectively). Error is depicted as S.E.M.; p-value determination was done via a Wilcoxon–Mann–Whitney test. P-value definitions are as shown in panel (B). Statistical data plotting and analysis was performed using the “Estimation Statistics” online tool [1].

1. Ho, Jose, et al. "Moving beyond P values: data analysis with estimation graphics." *Nature methods* 16.7 (2019): 565-566.

Multiple figures (e.g. Fig. 4) are missing comparison bars for matching p-values to comparisons on the graph. Finally, I believe the Nature standard is to show the individual data points for bar graphs where possible, and it is possible for many graphs in the manuscript.

We have updated the main text Figures 3, 4 and 6 and supplementary figures - Fig.S13, Fig. S18- Fig.S20, Fig. S22 to show the individual datapoints.

4. The excited state lifetime is extraordinary long ($\tau_1=14$ ps and $\tau_2=75$ ps in Archon1 vs 0.3 ps in Arch3). Taking this into account we did not observe a time component that would correlate to the observed kinetic voltage response ($\tau_f=0.61$ ms, $\tau_s=8.1$ ms ...).

The authors note the difference between the excited state and the voltage response kinetics. However, there are multiple components of the photocycle that are substantially longer than 1 ms in the Arch family, such as the M, N, or O states (Dougal Maclaurin et al, PNA, 110 5939). Could the authors please put the sensor kinetics into context of these other states?

This remark introduces the question - What is voltage response and voltage response kinetics? This has been a long standing question in the rhodopsin community and has been specifically addressed in our manuscript.

In case of light activated proton transporters, like Arch3, and model protein Bacteriorhodopsin a photocycle with consequent K/M/N/O photo-intermediates drive the proton translocation across cell membrane. The vectorial proton transport is voltage

dependent, where the transition between M-state and N-state is suggested to be voltage dependent (Arch3 - Maclaurin *et al. PNAS* 2013 Fig. 2E; BR – Geibel *et al. Biophysical Journal* 2001). In this study we observe voltage dependent reorientation of Arg92 in Arch3, which then might drive the voltage dependent pumping. The role of Arg82 flipping in BR (analogue to Arg92 in Arch3) proton transport has been discussed previously (Hutson *et al. Biochemistry* 2000, Clemens *et al. J. Phys. Chem.* 2011).

However in case of QuasArs and Archons, we no longer observe a typical photocycle, we no longer have a relaxation along consequent K/M/N intermediates. Upon excitation we have long lived excited state, but the chromophore isomerisation is almost completely blocked. This results in an increased fluorescent quantum yield of the microbial rhodopsin. There is only a minor fraction of molecules (Quasar1 – 1%, Archons < 0.1%) that isomerise, where most return back to original dark state. Moreover, the HPLC and spectroscopic results suggest a complex isomerization scheme (main text Fig.7a), where red light drives both single and double isomerization. This leads to a heterogeneous mixture of spectroscopically overlapping protonated species with all-*trans*/15-*anti* and 13-*cis*/15-*syn* retinal isomers. Discussed in detail in the main text discussion lines 520 – 567.

From the experimental studies and the MD simulations, our conclusion is that the voltage response is based on the reorientation of the side groups in the molecule, which causes alteration in the interaction between counterion and Schiff base. We do not know anything about the kinetics of the rearrangement after the voltage is changed and that is beyond our focus in the current study.

The phenomenon named “sensor kinetics” would correspond to the conversion of the protonated state (both all-*trans*/15-*anti* and 13-*cis*/15-*syn* retinal isomers species) into the deprotonated states. The voltage sensitive deprotonation kinetics and efficiency are expected to differ for between the retinal isomers, and have no correlation with the “photocycle kinetics” of the parental light driven pumps. Furthermore, as we pointed out in our previous correspondence to the reviewer - although the voltage sensitivity of Archon1 is observed under illumination, we believe that the voltage-dependent proton oscillation between the RSBH⁺ and the proton acceptor is light independent.

5. Taking into consideration the complex nature of D95E mutation we included a supplementary Fig. S30 (Figure R8) to demonstrates the lack of pumping activity in Archon1-Q95E-T99C (ARies1).

We appreciate the authors’ addition of this analysis. However, the comparison to WT Arch is not the most illustrative, as most D95X mutations will substantially decrease photocurrent. However, there are still cases with small residual photocurrent, which is not clearly shown on the new figure. These small photocurrents could still perturb the system in neuroscience studies, for example by activating/inhibiting neurons that are slightly below/above threshold. Please adjust the axes to zoom in on these reduced photocurrents. The n is also very low for these new experiments, but please provide statistical analyses on whether these values are non-zero or not, or a clarification that the presented data do not offer any conclusion on whether the photocurrent is non-zero.

We provide an additional panel (b) in Fig.S30 (here – Fig.R3) to show the raw recordings of photocurrents, that do not indicate any residual photocurrents. There is no indication of any peak or stationary currents, and we cannot carry out statistics on zero current.

Additionally, the focus of the current work has been to understand the underlying mechanism of the microbial rhodopsin based fluorescent voltage sensors and the development of tools for neuroscientists goes beyond the scope of this study.

Fig. R3 Comparison of photocurrent recordings for Arch3, Archon1 counterion mutants Q95H and Q95E T99C

a) Comparison of exemplary photocurrents for Arch3 and Archon1 derivatives. b) Zoom-in on recorded photocurrents of Archon1, Archon1-Q95H and Aries. c) Upper panel: Normalized stationary photocurrents at the denoted excitation wavelengths for Arch3 WT, normalized to photocurrents at 550 nm excitation and 0 mV holding potential (N=5). Lower panel: Photocurrent densities for Arch3 WT (N=5), Archon1 (N=3), Archon1 Q95H (N=3) and ARies1 (N=3). d) Photocurrent densities for all variants under all conditions tested.

REVIEWERS' COMMENTS

Reviewer #3 (Remarks to the Author):

The authors Silapetere present a revised manuscript about the photophysical properties of rhodopsin voltage indicators. Given the updated results, their work warrants publication. There are a couple of minor quibbles, and the authors may update their work as an option:

1. The Maclaurin PNAS 2013 paper showed that fluorescence changes non-linearly with excitation power for some versions of Arch. It is possible that this non-linearity plays some role in non-linear dF/F response with excitation power.
2. While the authors updated Fig. S30, it could be informative to show the photocurrent for the variants with small current on a different pA/pF axis. This would more clearly show the mean and variance of those measurements relative to zero.

Reviewers comments in **black**, answers from the authors in **blue**

Reviewer #3 (Remarks to the Author):

The authors Silapetere present a revised manuscript about the photophysical properties of rhodopsin voltage indicators. Given the updated results, their work warrants publication. There are a couple of minor quibbles, and the authors may update their work as an option:

1. The Maclaurin PNAS 2013 paper showed that fluorescence changes non-linearly with excitation power for some versions of Arch. It is possible that this non-linearity plays some role in non-linear dF/F response with excitation power.

The observations in Maclaurin et al. 2013 *PNAS* study (e.g. Arch3 wild-type shown in Fig. 2c) can be explained with fluorescence arising from a multi-photon process (Fig.3D). The first photon initiates a photocycle typical for microbial-rhodopsins. The second photon absorbed by the N-state initiates photo-branching and brings the molecule to the fluorescent Q-state. Ultimately, the third photon absorbed by the Q-state is re-emitted. This results in non-linear dependency of fluorescence intensity and excitation intensity.

However in the case for QuasArs and Archons fluorescence arises from a single photon excitation. For these constructs the dark state is the fluorescent state - single photon excitation results in extraordinary long lived excited state and increased fluorescence quantum yield (compared to Arch3 wild-type). This results in linear dependence between excitation intensity and fluorescence. Comparison between QuasArs and Arch3 is shown in Hochbaum et al. 2014 *Nat. Comm.* (Fig. 1b).

2. While the authors updated Fig. S30, it could be informative to show the photocurrent for the variants with small current on a different pA/pF axis. This would more clearly show the mean and variance of those measurements relative to zero.

The supplementary Figure S30 has been updated according to reviewers #3 suggestion.